# PARETO-CONDITIONED DIFFUSION MODELS FOR OFFLINE MULTI-OBJECTIVE OPTIMIZATION

**Jatan Shrestha***
Aalto University

**Santeri Heiskanen***
Aalto University

**Kari Hepola**
Tampere University

**Severi Rissanen**
Aalto University

**Pekka Jääskeläinen**
Tampere University

**Joni Pajarinen**
Aalto University

## ABSTRACT

Multi-objective optimization (MOO) arises in many real-world applications where trade-offs between competing objectives must be carefully balanced. In the offline setting, where only a static dataset is available, the main challenge is generalizing beyond observed data. We introduce Pareto-Conditioned Diffusion (PCD), a novel framework that formulates offline MOO as a conditional sampling problem. By conditioning directly on desired trade-offs, PCD avoids the need for explicit surrogate models. To effectively explore the Pareto front, PCD employs a reweighting strategy that focuses on high-performing samples and a reference-direction mechanism to guide sampling towards novel, promising regions beyond the training data. Experiments on standard offline MOO benchmarks show that PCD achieves highly competitive performance and, importantly, demonstrates greater consistency across diverse tasks than existing offline MOO approaches.

## 1 INTRODUCTION

Multi-objective optimization (MOO) tackles real-world problems involving conflicting objectives (Miettinen, 1998; Deb, 2001). Such scenarios arise in domains from scientific discovery to engineering, where performance must be balanced against cost and energy. In many of these domains, evaluating candidate designs is prohibitively expensive, time-consuming, or even risky. For instance, in biological sequence design, each experiment or simulation incurs substantial cost (Angermueller et al., 2020). In such settings, online optimization methods, which rely on repeatedly querying a black-box objective function, are often impractical (Trabucco et al., 2022). This challenge has motivated the development of offline MOO, which instead relies on a pre-collected dataset of previously evaluated designs, without access to the true objective during optimization (Xue et al., 2024).

The offline setting presents unique and significant challenges. Unlike online methods, algorithms cannot query the true objective function during training and are restricted to a static dataset (Trabucco et al., 2022). This constraint introduces a fundamental tradeoff as the algorithm must remain faithful to the data to avoid unreliable predictions, yet also explore beyond the observed designs to identify superior solutions (Trabucco et al., 2022). Moreover, offline MOO requires finding a set of solutions that balances multiple objectives, rather than a single optimum (Xue et al., 2024). This leads to a particularly difficult form of generalization: algorithms must be creative enough to find solutions that improve upon the training data, while being conservative enough to stay within the reliable boundaries of the training data manifold (Trabucco et al., 2022; Xue et al., 2024).

The most dominant approach to offline MOO is model-based optimization (Kim et al., 2026), typically using a surrogate model (Trabucco et al., 2021; Yu et al., 2021; Yuan et al., 2023b; Qi et al., 2022; Chen et al., 2023; Daulton et al., 2020; 2021). The surrogate is trained on the static dataset to approximate the black-box objective and then used with multi-objective evolutionary algorithms (MOEAs) (Deb, 2001; Deb et al., 2002) to search for promising solutions. More recent approaches use generative models to directly synthesize solutions (Kim et al., 2023). For instance, ParetoFlow

---

*Equal contribution. Correspondence to: `jatan.shrestha@aalto.fi`, `santeri.heiskanen@aalto.fi`
Project website: https://sites.google.com/view/pcd-iclr26

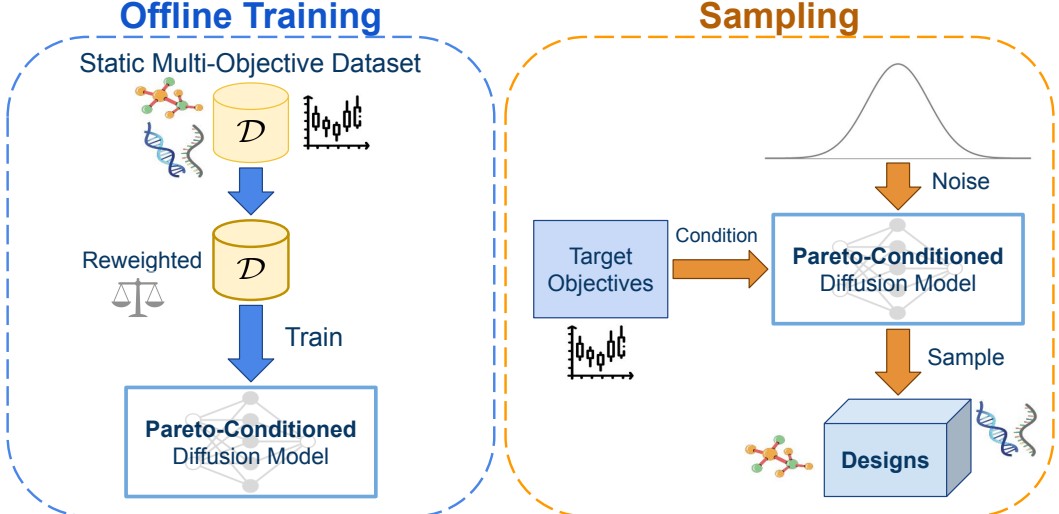

Figure 1: **Overview of the PCD framework, which reframes offline MOO as a conditional sampling problem. Training:** A conditional diffusion model is trained on a static dataset, using a novel reweighting strategy to emphasize high-quality solutions near the Pareto front. **Sampling:** At inference, the model directly generates novel designs conditioned on target objectives. This end-to-end approach sidesteps the need for the explicit surrogate models and separate optimizers required by prior methods.

(Yuan et al., 2025) employs flow matching-based generative models guided by a multi-objective predictor to generate samples along the Pareto front. However, these methods still rely on surrogate predictors to guide the generation process, inheriting the same potential risks of exploiting inaccuracies in the surrogate model (Trabucco et al., 2021).

**Contribution.** In this work, we propose *Pareto-Conditioned Diffusion (PCD)*, a novel framework that reframes offline MOO as a conditional sampling problem, enabling the generation of high-quality solutions conditioned directly on target trade-offs (see Fig. 1). Unlike prior methods (Xue et al., 2024; Yuan et al., 2025) that rely on multi-stage pipelines of surrogate predictors or separate optimization algorithms, PCD offers a more direct, end-to-end framework. It unifies the generation of candidate solutions and the modeling of the Pareto front into a single conditional generative model, simplifying the overall optimization process without requiring explicit objective function approximation.

To enable this framework, we introduce two key algorithmic contributions: (1) a multi-objective reweighting strategy that biases the model to focus on high-performing regions of the data distribution, and (2) a reference-direction mechanism to generate diverse and high-quality conditioning points that enable generalization beyond the observed dataset. We provide extensive empirical validation on a wide range of standard offline MOO benchmarks. These experiments demonstrate that PCD achieves highly competitive performance and, more importantly, exhibits significantly greater consistency across a wide variety of tasks than existing offline MOO approaches. The effectiveness of our proposed reweighting and conditioning mechanisms is confirmed through rigorous ablation studies.

## 2 RELATED WORK

**Offline Multi-Objective Optimization.** Prior work in offline MOO has primarily focused on model-based approaches (Xue et al., 2024; Kim et al., 2026). These methods train a surrogate model on a static dataset to approximate the black-box objective function. Surrogates can take the form of Deep Neural Networks (DNNs) (Trabucco et al., 2021; Yu et al., 2021; Qi et al., 2022; Yuan et al., 2023b; Chen et al., 2023), which scale well to high-dimensional search spaces, or Gaussian Processes (GPs) (Daulton et al., 2020; 2021), which provide uncertainty estimates useful for explo-

ration. Once the surrogate is trained, a separate search algorithm, most commonly MOEAs such as NSGA-II (Deb, 2001; Deb et al., 2002), is used to optimize the learned proxy to find promising solutions (Xue et al., 2024). The performance of this multi-stage pipeline, however, is fundamentally bottlenecked by the accuracy of the surrogate model (Trabucco et al., 2021).

**Generative Modeling Methods.** More recent approaches use generative models to directly synthesize solutions (Kim et al., 2023). The primary advantage of these methods is their ability to generate diverse designs without relying on a separate, explicit search algorithm (Kim et al., 2026). For instance, ParetoFlow (Yuan et al., 2025) and PGD-MOO (Annadani et al., 2025) employ generative models guided by a multi-objective predictor. Similarly, LaMBO (Stanton et al., 2022) integrates Bayesian optimization to steer its diffusion model, while MOGFNs (Jain et al., 2023) utilize generative flow networks often steered by scalarization functions. However, the effectiveness of these methods is still dependent on a surrogate predictor or a scalarization function to guide the generator (Yuan et al., 2025; Annadani et al., 2025).

In contrast, our method, PCD, removes this dependency on intermediate guidance mechanisms like explicit surrogate models or scalarization. PCD frames offline MOO as a conditional sampling problem, generating high-quality solutions by conditioning the diffusion model directly on target objectives. In doing so, PCD extends prior work on single-objective conditional generation, such as Model Inversion Networks (MINs) (Kumar & Levine, 2020) and Denoising Diffusion Optimization Models (DDOM) (Krishnamoorthy et al., 2023), to the more complex multi-objective setting. Our work is also distinct from reward-directed diffusion models (Yuan et al., 2023a), which focus on maximizing a single reward signal rather than balancing a set of conflicting objectives.

## 3 BACKGROUND

**Offline Multi-Objective Optimization.** Let $\boldsymbol{f} : \mathcal{X} \to \mathbb{R}^m$ be a vector-valued function that maps a solution from the solution space $\mathcal{X}$ to the $m$-dimensional objective space $\mathbb{R}^m$. A multi-objective optimization (MOO) problem can be formally defined as

$$\min_{\boldsymbol{x} \in \mathcal{X}} \boldsymbol{f}(\boldsymbol{x}) = [f_1(\boldsymbol{x}), \ldots, f_m(\boldsymbol{x})], \tag{1}$$

where $\boldsymbol{x} \in \mathcal{X} \subseteq \mathbb{R}^d$ is a solution vector of dimension $d$, and $\boldsymbol{f}(\boldsymbol{x})$ consists of $m$ objective functions that must be minimized simultaneously (Miettinen, 1998; Deb, 2001). Since there is generally no single solution $\boldsymbol{x}^*$ that minimizes all objectives simultaneously, solutions are compared using Pareto dominance.

**Definition 1** (Pareto Dominance). *An objective vector $\boldsymbol{f}(\boldsymbol{x})$ is said to Pareto-dominate another vector $\boldsymbol{f}(\boldsymbol{x}')$, denoted $\boldsymbol{f}(\boldsymbol{x}) \prec \boldsymbol{f}(\boldsymbol{x}')$, if $\forall i : f_i(\boldsymbol{x}) \leq f_i(\boldsymbol{x}')$ with at least one strict inequality.*

**Definition 2** (Pareto Optimality). *A solution $\boldsymbol{x}^* \in \mathcal{X}$ is called Pareto-optimal if no other solution $\boldsymbol{x} \in \mathcal{X}$ Pareto-dominates it. The set of all such solutions is the Pareto-optimal set (PS) and their corresponding set of objective vectors, $\{\boldsymbol{f}(\boldsymbol{x}^*) \mid \boldsymbol{x}^* \in PS\}$, forms the Pareto front (PF).*

The goal of MOO is to approximate the PF with a diverse and well-distributed set of solutions. When evaluating the objective function $\boldsymbol{f}$ is expensive, as in drug discovery or architecture search, offline MOO uses a pre-collected dataset $\mathcal{D} = \{(\boldsymbol{x}_i, \boldsymbol{y}_i)\}_{i=1}^N$, with $\boldsymbol{y}_i = \boldsymbol{f}(\boldsymbol{x}_i)$, without direct access to $\boldsymbol{f}$ during training (Xue et al., 2024; Kim et al., 2026). Offline methods rely solely on this static dataset $\mathcal{D}$, though a limited budget $Q$ of new queries to $\boldsymbol{f}$ may be allowed during evaluation. Solution quality is typically measured by the Hypervolume (HV) indicator, which is calculated relative to a reference point derived from the best-known solutions when the true PF is unknown (Zitzler & Thiele, 1998; Zitzler et al., 2007).

**Conditional Diffusion Models.** Diffusion models are a powerful class of generative models that produce high-quality and diverse samples by iteratively denoising corrupted data (Ho et al., 2020; Song et al., 2021b). Building on the Elucidated Diffusion Model (EDM) formulation (Karras et al., 2022), we extend it to the conditional setting. The clean data is drawn from a joint distribution $(\boldsymbol{x}, \boldsymbol{y}) \sim p_{\text{data}}(\boldsymbol{x}, \boldsymbol{y})$, and the conditional smoothed distribution is defined as $p(\boldsymbol{x} \mid \boldsymbol{y}; \sigma)$, obtained by adding Gaussian noise of standard deviation $\sigma$ to $\boldsymbol{x}$. The evolution of a noisy conditional sample

$\boldsymbol{x} \sim p(\boldsymbol{x} \mid \boldsymbol{y}; \sigma)$ as the noise level decreases is described by the ordinary differential equation (ODE):

$$\mathrm{d}\boldsymbol{x}/\mathrm{d}\sigma = -\big(D_\theta(\boldsymbol{x}; \boldsymbol{y}, \sigma) - \boldsymbol{x}\big)/\sigma. \tag{2}$$

The denoiser network $D_\theta$ is trained to minimize the expected $L_2$ denoising error:

$$\theta = \arg\min_\theta \mathbb{E}_{(\boldsymbol{x},\boldsymbol{y}) \sim p_{\text{data}}, \sigma \sim p_{\text{train}}, \boldsymbol{n} \sim \mathcal{N}(0,\sigma^2 \mathbb{I})} \lambda(\sigma) \|D_\theta(\boldsymbol{x} + \boldsymbol{n}; \boldsymbol{y}, \sigma) - \boldsymbol{x}\|_2^2. \tag{3}$$

Here, $p_{\text{train}}(\sigma)$ specifies the noise distribution used during training and $\lambda(\sigma)$ is a weighting factor between noise levels. Sampling starts from pure noise $\boldsymbol{x}_0 \sim p(\boldsymbol{x} \mid \boldsymbol{y}; \sigma_{\text{max}})$ and follows the ODE down to $\sigma = 0$, yielding a sample from the conditional data distribution.

To steer generation towards a desired conditioning target $\boldsymbol{y}$, we employ classifier-free guidance (CFG) (Ho & Salimans, 2022). This modifies the ODE by combining a conditional denoiser $D_\theta(\boldsymbol{x}; \boldsymbol{y}, \sigma)$ with an unconditional denoiser $D_\theta(\boldsymbol{x}; \sigma)$, where the unconditional denoiser is implemented using the same network by randomly dropping conditioning during training. The modified ODE is:

$$\mathrm{d}\boldsymbol{x}/\mathrm{d}\sigma = -(\gamma D_\theta(\boldsymbol{x}; \boldsymbol{y}, \sigma) + (1-\gamma)D_\theta(\boldsymbol{x}; \sigma) - \boldsymbol{x})/\sigma, \tag{4}$$

where $\gamma$ controls the guidance strength. Setting $\gamma > 1$ amplifies the influence of the conditioning target, guiding the samples to regions consistent with $\boldsymbol{y}$.

## 4 METHOD

Building on the conditional diffusion framework, PCD reframes offline MOO as a conditional generation problem, where $\boldsymbol{x}$ represents candidate solutions and the conditioning vector $\boldsymbol{y}$ corresponds to the target objectives. The diffusion model learns to directly model the conditional distribution $p(\boldsymbol{x} \mid \boldsymbol{y}; \sigma)$ of solutions $\boldsymbol{x}$ given objectives $\boldsymbol{y}$. In the following sections, we describe the two key components that enable this approach: (1) a reweighting strategy to emphasize high-quality designs near the Pareto front, and (2) a reference-direction mechanism to steer sampling toward preferred trade-offs and generalize beyond the observed dataset.

**Training via Multi-Objective Reweighting.** As stated, the goal of offline MOO is to discover new solutions that outperform those in the offline dataset $\mathcal{D}$. To achieve this, the model must generalize accurately near well-performing samples, while modeling of low-performing regions is less important. A natural way to do this is to place higher emphasis on high-performance points by reweighting the dataset (Kumar & Levine, 2020; Krishnamoorthy et al., 2023). Simply pruning poorly performing points, as in (Xue et al., 2024), can increase the variance of the training objective (Kumar & Levine, 2020; Krishnamoorthy et al., 2023).

A key challenge when reweighting the offline dataset is how to rank the samples. As noted earlier, there is no total ordering between $\boldsymbol{f}(\boldsymbol{x})$ and $\boldsymbol{f}(\boldsymbol{x}')$, so simple strategies, such as measuring the distance to the best point in the dataset, are not feasible. Instead, points can be ranked based on the number of points that dominate them. Specifically, one can compute the *dominance number* (Zhao et al., 2022):

$$o(\boldsymbol{x}) := \sum_{\boldsymbol{x}' \in \mathcal{D}} \mathbb{I}[\boldsymbol{f}(\boldsymbol{x}) \prec \boldsymbol{f}(\boldsymbol{x}'), \boldsymbol{x} \neq \boldsymbol{x}'] \tag{5}$$

where $\mathbb{I}$ is the indicator function. Intuitively, a lower dominance number indicates that a point performs better relative to other points in the offline dataset.

To reweight the dataset, the objective space $\mathbb{R}^m$ is first partitioned into a grid of $N_B$ bins of equal width. Points in a bin are then weighted to emphasize bins that 1) contain more points, and 2) are well-performing on average. We implement this using the following multi-objective reweighting scheme, similar to (Kumar & Levine, 2020; Krishnamoorthy et al., 2023):

$$w_i = \frac{|B_i|}{|B_i| + K} \exp\Big(\frac{-\frac{1}{|B_i|}\sum_{j=1}^{|B_i|} o(\boldsymbol{x}_{b_j})}{\tau}\Big) \tag{6}$$

where $|B_i|$ is the size of the $i$-th bin, $\boldsymbol{x}_{b_j}$ is the $j$-th sample of $i$-th bin and $K, \tau$ are hyperparameters controlling smoothing over bin sizes and the temperature of the weighting distribution. The first

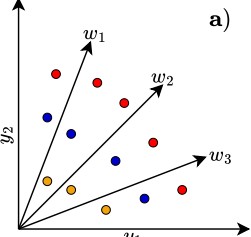 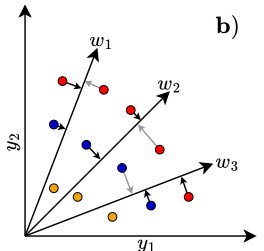 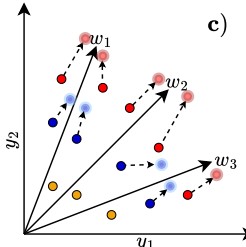

Figure 2: **Overview of the conditioning points generation procedure**: **a)** The objective space is partitioned via direction vectors, and points are ranked based on non-dominated sorting. **b)** Each direction vector is paired with the point closest to it in perpendicular distance (black arrow). The rest of the points are paired to the vector with the least amount of assigned points (gray arrow). **c)** A diverse set of conditioning points is generated by extrapolating the assigned points along the direction vectors and adding Gaussian noise.

term emphasizes bins with more points, while the second assigns higher weight to bins containing high-performance points. Therefore, Equation 6 achieves both desired properties. Appendix A.2 discusses practical implications of reweighting in more detail.

To put it all together, Equation 6 is applied to Equation 3, resulting in the reweighted denoising $L_2$ objective:

$$\theta = \arg\min_{\theta} \mathbb{E}_{(\boldsymbol{x},\boldsymbol{y})\sim p_{\text{data}},\, \sigma \sim p_{\text{train}},\, \boldsymbol{n}\sim\mathcal{N}(0,\sigma^2\mathbb{I})} w(\boldsymbol{y})\,\lambda(\sigma)\,\|D_\theta(\boldsymbol{x}+\boldsymbol{n};\boldsymbol{y},\sigma)-\boldsymbol{x}\|_2^2, \qquad (7)$$

where $w(\boldsymbol{y})$ is the reweighting factor from Equation 6, and $\lambda(\sigma)$ is the noise-level-dependent weighting term.

**Generating a diverse set of conditioning points.** A crucial question in conditional sampling is how to select the target conditioning points, which we denote as $\hat{\boldsymbol{y}}$. While simple tactics exist for single-objective optimization (Kumar & Levine, 2020), they are ineffective for MOO where the goal is to approximate the entire Pareto front. We therefore propose a two-stage procedure to generate a set of $Q$ conditioning points $\hat{\boldsymbol{y}}$, designed to be both high-quality and diverse. The procedure is described briefly below, with full pseudocode and explanation in Appendix A.9.

To collect the set of representative points, we propose to use a procedure motivated by the survival method in NSGA-III (Deb & Jain, 2014). Firstly, the objective space is partitioned by generating $L$ direction vectors $\boldsymbol{w}_i$ via Riesz s-Energy method (Hardin & Saff, 2005). Secondly, the points in the offline dataset are ranked via non-dominated sorting, resulting in a list of fronts $(F_1, F_2, \ldots, F_M)$, where each front consists of a set of points with the same rank. Lastly, we iterate over the fronts to assign each point a direction vector $\boldsymbol{w}_i$. Initially, each direction vector gets assigned a point that is closest to it in perpendicular distance. Then, the rest of points in front $F_i$ are iterated over one-by-one and assigned to a direction vector with the least amount of points at the moment. This procedure is continued until $Q$ (point, direction vector) pairs have been created.

After collecting the set of representative points, the points are extrapolated towards the assigned direction vector to improve their quality over the points in the offline dataset. However, in practice the number of solutions $Q$ is often larger than the number of direction vectors $L$. Therefore multiple points get extrapolated towards the same direction, which decreases the diversity of the conditioning points. To mitigate this, the final conditioning points are obtained by adding zero-mean Gaussian noise to the extrapolated points, akin to the Gaussian mutation operator used in evolutionary algorithms. This reweighted objective, paired with reference direction sampling, forms the core of the proposed PCD.

**Sampling with Classifier-Free Guidance.** For each conditioning point $\hat{\boldsymbol{y}}$ generated by our reference-direction mechanism, we sample a corresponding solution $\boldsymbol{x}$ by applying Classifier-Free Guidance (CFG) (Ho & Salimans, 2022). As formally defined in Section 3, CFG steers the denoising process by creating a weighted combination of the model's general and target-specific predictions.

---

**Algorithm 1** PCD Training and Sampling

---

**Input:** Dataset $\mathcal{D} = \{(\boldsymbol{x}_i, \boldsymbol{y}_i)\}_{i=1}^N$, hyperparameters for reweighting $(N_B, K, \tau)$ and sampling $(Q, L, \gamma, S, \sigma_{\min}, \sigma_{\max})$
**Output:** Candidate set $\boldsymbol{X}$ with $|\boldsymbol{X}| = Q$
**Phase 1: Training**
1: Initialize model parameters $\theta$
2: Train denoiser $D_\theta$ on $\mathcal{D}$ using the reweighted objective (Eq. 7)
   **Phase 2: Diffusion Sampling**
3: Generate conditioning points $\hat{\boldsymbol{Y}} = \{\hat{\boldsymbol{y}}_i\}_{i=1}^Q$ using Algorithm 2
4: $\boldsymbol{X} \leftarrow \emptyset$
5: **for** each $\hat{\boldsymbol{y}} \in \hat{\boldsymbol{Y}}$ **do**
6:      Sample $\boldsymbol{x} \sim \mathcal{N}(\boldsymbol{0}, \sigma_{\max}^2 \mathbb{I})$            ▷ Start sampling from pure noise
7:      **for** $\sigma$ from $\sigma_{\max}$ to $\sigma_{\min}$ **over** $S$ **steps do**
8:          $\hat{D}_\theta(\boldsymbol{x}; \hat{\boldsymbol{y}}, \sigma) \leftarrow \gamma D_\theta(\boldsymbol{x}; \hat{\boldsymbol{y}}, \sigma) + (1 - \gamma) D_\theta(\boldsymbol{x}; \sigma)$    ▷ Apply Classifier-Free Guidance
9:          Update $\boldsymbol{x} \leftarrow \texttt{ODE-STEP}(\boldsymbol{x}, \hat{D}_\theta, \sigma)$        ▷ Perform one step of denoising
10:      **end for**
11:      $\boldsymbol{X} \leftarrow \boldsymbol{X} \cup \{\boldsymbol{x}\}$
12: **end for**
13: **return** $\boldsymbol{X}$

---

This well-established technique, widely used for high-fidelity image generation (Saharia et al., 2022; Rombach et al., 2022), strongly enforces adherence to the conditioning signal. In our setting, this biases the generated sample $\boldsymbol{x}$ to be highly faithful to the target trade-off $\hat{\boldsymbol{y}}$, yielding a final set of solutions that accurately covers the desired region of the Pareto front. The complete training and sampling procedure is summarized in Algorithm 1.

## 5 EXPERIMENTAL VALIDATION

We conducted extensive experiments to validate the performance of the proposed method against a variety of baselines from the literature. In Section 5.1, we give an overview of the used benchmark, while Section 5.2 introduces the baselines. Section 5.3 discusses the main results, and Section 5.4 validates the performance of the proposed components via ablation studies.

### 5.1 OVERVIEW

Our proposed approach is evaluated using the comprehensive Offline MOO benchmark (Xue et al., 2024). We consider five diverse task categories: 1) synthetic, 2) multi-objective reinforcement learning (MORL), 3) real-world applications (RE), 4) scientific design, and 5) multi-objective neural architecture search (MONAS), each of which is described in more detail below. For benchmark tasks that involve discrete values, we convert them into continuous logits, following the standard practice in prior work (Trabucco et al., 2022; Xue et al., 2024).

**Tasks.** The synthetic task set consists of 11 commonly used multi-objective test functions, including DTLZ1, DTLZ7[1], OmniTest, VLMOP1-3 and ZDT1-4, ZDT6. These are all continuous-valued functions with datasets of 60,000 points each, featuring two or three objectives with varying Pareto front shapes.

The MORL task set contains two continuous high-dimensional problems, MO-Hopper and MO-Swimmer, each with two objectives, where the goal is to generate optimal parameters for a neural network control policy. The offline datasets are composed of policy parameters collected during PGMORL training (Xu et al., 2020).

The Real-World Applications (RE) task set consists of engineering design problems, such as pressure vessel and rocket injector design (Tanabe & Ishibuchi, 2020). These tasks have 4-7 dimen-

---

[1]Tasks DTLZ2-DTLZ6 are excluded due to evaluation errors in the original benchmark: `https://github.com/lamda-bbo/offline-moo/issues/14`

sional search spaces and 2-6 objectives. Unlike the previous task sets, several RE tasks involve mixed-integer search spaces.

The Scientific Design task set includes one molecule design problem and three discrete protein design problems. The molecule task is optimized in a 32-dimensional latent space, while the protein tasks involve generating sequences of 32 to 200 tokens to optimize two conflicting objectives.

The MONAS task set aims to optimize the accuracy, latency, and parameter count of a neural network. With up to 34 decision variables, this is the only purely categorical task set in our evaluation, posing a unique challenge for continuous generative models.

**Excluded Tasks.** While the original benchmark suite by (Xue et al., 2024) included combinatorial tasks, we exclude them here. Adapting diffusion models to combinatorial domains requires substantial modifications to the denoising process and decoding strategy (Sun & Yang, 2023; Sanokowski et al., 2024), which is beyond our current scope. Following prior work (Yuan et al., 2025), we focus on continuous, discrete and categorical tasks, which we believe sufficiently illustrate the strengths and limitations of our proposed method.

**Setup.** The quality of the generated solution set is measured using the Hypervolume (HV) indicator, following the original benchmark (Xue et al., 2024). Formally, HV measures the volume of the objective space that is weakly dominated by a solution set (Zitzler & Thiele, 1998). HV is a comprehensive metric that condenses the quality of a set into a single number while simultaneously measuring its 1) uniformity, 2) spread, and 3) convergence (Zintgraf et al., 2015), making it ideal for comparing methods. Following prior work (Trabucco et al., 2022; Xue et al., 2024), we evaluate performance on different subsets of the generated solutions. Specifically, we first generate a set of $Q = 256$ candidate solutions. We then report the HV on the top $P$ percentile of this set for $P \in \{100, 75, 50\}$. For $P < 100$, the lowest-performing $(100 - P)\%$ of solutions are removed before HV calculation. All results are reported as the mean and standard deviation over 5 random seeds unless specified otherwise. Detailed hyperparameters for our method PCD and training procedure are provided in Appendix A.3.

## 5.2 BASELINES

Our proposed method is compared against a comprehensive set of baselines, primarily drawn from the Off-MOO-Bench benchmark (Xue et al., 2024). The majority of these methods are *surrogate-based*: they first train a model on the offline data to approximate the objective functions, and then use this learned proxy to guide a search algorithm. These baselines primarily differ in the type of surrogate model used, for which we consider both DNN-based and GP-based approaches.

*DNN-based surrogates* are categorized into three subclasses based on their architecture. *End-to-end (E2E)* models use a single DNN to output an $m$-dimensional vector representing all objectives. In contrast, *Multiple Models (MM)* maintain a separate regressor for each of the $m$ objectives. As each model is trained on a single objective, this approach allows for pairing with powerful offline single-objective optimization techniques such as COMs (Trabucco et al., 2021), ICT (Yuan et al., 2023b), IOM (Qi et al., 2022), and Tri-Mentoring (TM) (Chen et al., 2023). Finally, inspired by multi-task learning, *Multi-Head (MH)* models use a shared feature encoder with separate output heads for each objective, enabling efficient information sharing. MH models are often paired with training techniques like PCGrad (PC) (Yu et al., 2020) and GradNorm (GN) (Chen et al., 2018) to improve generalization.

For *GP-based surrogates*, we only consider the prominent qNEHVI algorithm (Daulton et al., 2020). Following the findings of (Xue et al., 2024), we omit other GP models for two reasons: 1) training them on large offline datasets is computationally expensive, and 2) DNN-based models have been shown to outperform them on most offline MOO tasks. Regardless of the surrogate type (DNN or GP), the final solution set is generated using the NSGA-II search algorithm (Deb et al., 2002) with a budget of $Q = 256$ evaluations on the true oracle.

Finally, we include *generative model baselines*. ParetoFlow (Yuan et al., 2025) is the only other generative model *specifically* developed for offline MOO that we are aware of and is our primary

Table 1: Average rank ($\downarrow$) of PCD and baseline methods across five task categories. Ranks are calculated based on the 100th percentile HV. **Bold** and underlined rows indicate the best and runner up methods respectively. PCD achieves the best overall average rank, demonstrating its strong and consistent performance.

| Method | Synthetic | MORL | RE | Scientific | MONAS | Avg. rank |
|---|---|---|---|---|---|---|
| $\mathcal{D}$(best) | 5.45±0.19 | **1.70±0.27** | 2.60±0.07 | 9.35±0.14 | 11.53±0.06 | 7.43±0.05 |
| MOBO | 8.69±0.30 | 14.60±0.42 | 10.00±0.33 | 6.75±0.47 | 8.11±0.80 | 8.81±0.34 |
| E2E + GN | 7.33±0.55 | 5.70±2.14 | 7.06±0.32 | 5.35±1.38 | 9.33±0.53 | 7.82±0.40 |
| E2E + PC | 5.93±0.25 | 3.50±1.22 | 6.22±0.33 | 4.30±1.32 | 6.60±0.40 | 6.01±0.29 |
| E2E | 6.16±0.30 | 9.70±2.08 | 6.06±0.30 | 4.20±1.40 | 5.13±0.22 | 5.71±0.16 |
| MH + GN | 8.82±0.53 | 8.90±2.16 | 8.14±0.94 | 5.05±2.14 | 12.57±0.40 | 9.84±0.33 |
| MH + PC | 8.87±0.45 | 10.90±1.08 | 6.74±0.68 | 6.15±0.91 | 7.46±0.30 | 7.68±0.33 |
| MH | 6.18±0.53 | 8.00±1.41 | 6.14±0.29 | 5.80±0.89 | 5.88±0.49 | 6.10±0.22 |
| MM + COMs | 8.02±0.47 | 3.60±1.29 | 6.54±0.17 | **3.85±0.68** | 7.22±0.43 | 6.80±0.13 |
| MM + ICT | 6.73±0.46 | 9.10±1.95 | 5.44±0.32 | 5.05±0.74 | 8.42±0.40 | 7.08±0.13 |
| MM + IOM | 5.16±0.51 | 12.70±0.91 | 5.76±0.52 | 4.40±1.15 | 5.77±0.50 | 5.80±0.20 |
| MM + TM | 6.55±0.82 | 7.90±2.16 | 5.78±0.25 | 5.90±1.29 | 7.87±0.39 | 6.91±0.20 |
| MM | 6.07±0.50 | 9.50±0.79 | 5.94±0.41 | 6.55±0.93 | **4.97±0.46** | 5.80±0.21 |
| ParetoFlow | **2.44±0.28** | 8.50±1.32 | 1.74±0.17 | 9.05±0.27 | 11.19±0.52 | 6.74±0.23 |
| PCD (ours) | 3.38±0.20 | 5.50±3.30 | **1.51±0.13** | 4.05±0.33 | 7.54±0.50 | **4.80±0.30** |

comparison[2]. While LaMBO (Stanton et al., 2022) is a related diffusion-based approach, it was used to generate some of the benchmark datasets themselves. Including LaMBO as a baseline could therefore introduce bias, so it is excluded. We also include a simple but strong baseline, $\mathcal{D}$**(best)**, which represents the performance of the non-dominated set of solutions present in the original offline dataset.

## 5.3 RESULTS

Table 1 summarizes our primary experimental findings, showing the average rank of each method across the five task categories based on the 100th percentile HV. The table also presents the total average rank aggregated over all tasks. For detailed per-task HV scores at the $P \in \{100, 75, 50\}$ percentiles, please see Appendix A.11, A.12, and A.13 respectively.

The results in Table 1 clearly demonstrate the effectiveness of our approach. **PCD achieves the best overall rank**, highlighting its consistent high performance across a diverse set of problems. This consistency is particularly noteworthy, as these results were obtained using a single, fixed set of hyperparameters, underscoring the method's robustness in the offline setting where such tuning is inherently difficult. Even in the MORL and MONAS task sets, where its performance is weaker relative to other categories, PCD still outperforms the other generative baseline, ParetoFlow (Yuan et al., 2025), by a considerable margin.

The relatively weaker performance on *MORL* tasks can be attributed to the extremely high-dimensional search space. In fact, this benchmark proved challenging for all evaluated methods. Notably, and in contrast to some prior reports (Yuan et al., 2025), *we found that no method could produce a solution set superior to the non-dominated points already present in the offline dataset*[3]. This finding highlights the immense difficulty of the MORL tasks and underscores the need for further research in this area.

Similarly, the performance on *MONAS* is impacted by its purely categorical search space, which presents a challenge for our continuous diffusion model. While diffusion models have achieved excellent results on discrete and graph-based data, doing so typically requires specialized architectural

---

[2]PGD-MOO (Annadani et al., 2025) is omitted because the results in the original study were limited to synthetic tasks, and we were unable to obtain results on our broader benchmarks using the official code.

[3]We hypothesize that the difference in our results and those of prior work may be due to updated datasets released by (Xue et al., 2024).

Table 2: Average 100th percentile HV ($\uparrow$) on 5 representative tasks with different data processing and sampling strategies. **Bold** and underlined rows indicate the best and runner up methods respectively. The results validate the effectiveness of both our proposed reweighting and reference-direction mechanisms.

| Variant | ZDT2 | MO-Swimmer-v2 | RE34 | Regex | C10/MOP2 |
|---------|------|---------------|------|-------|----------|
| Ideal + N/A | 7.59±0.19 | 1.76±0.21 | 9.19±0.26 | **5.60±0.35** | 10.46±0.01 |
| $\mathcal{D}$(best) + N/A | 7.53±0.41 | 3.58±0.06 | 9.32±0.01 | 4.84±0.48 | 10.44±0.02 |
| Ref. Dir. + N/A | **7.89±0.19** | 3.53±0.11 | 10.11±0.03 | 5.55±0.25 | 10.47±0.01 |
| Ref. Dir. + Pruning | 5.64±0.13 | 3.63±0.07 | 10.16±0.03 | 4.20±0.64 | 10.55±0.08 |
| PCD | 6.25±0.06 | **3.69±0.11** | **10.17±0.04** | 4.80±0.87 | **10.59±0.04** |

changes not employed in this work (An et al., 2024; Niu et al., 2020; Jo et al., 2022). Integrating these architectural advances into the PCD framework is a promising direction for future work.

## 5.4 ABLATION STUDIES

**Ablations on PCD components.** We conduct an ablation study to validate the two primary components of PCD introduced in Section 4: the multi-objective reweighting scheme and the reference-direction mechanism for generating conditioning points. We compare three data processing techniques: 1) no processing (N/A), 2) simple pruning of low-quality points as in (Xue et al., 2024), and 3) our proposed data reweighting. Additionally, we compare three strategies for generating conditioning points: our proposed reference-direction (Ref. Dir.) mechanism, conditioning on the best set of points in the offline dataset ($\mathcal{D}$(best)), and a simpler baseline (Ideal) that randomly samples non-dominated points from the dataset and extrapolates them toward the ideal objective vector, a strategy common in offline single-objective optimization. Note that the full PCD method combines our proposed reweighting with the reference-direction mechanism.

The results in Table 2 confirm the effectiveness of both of our proposed components. First, using the *reference-direction mechanism* to select conditioning points consistently improves or maintains performance compared to both the simplistic "Ideal" and "$\mathcal{D}$(best)" strategies. The benefit is especially pronounced on the MO-Swimmer task, where our approach nearly doubles the HV compared to the "Ideal" strategy, highlighting the critical importance of generating high-quality and diverse conditioning points. Second, the table showcases that our proposed *reweighting strategy consistently and significantly outperforms the simple pruning* approach from (Xue et al., 2024). Interestingly, for the ZDT2 and Regex tasks, any form of data processing appears to harm model performance. We hypothesize this is due to the quality of the underlying datasets in these specific cases; when a dataset is less skewed (i.e., most points are already high-quality), aggressive reweighting or pruning can discard valuable training signal for minimal gain. As explained further in Appendix A.1, these tasks appear to be the exception, as we observe no similar performance degradation in other benchmarks.

**Effect of guidance scale.** Finally, we ablate the effect of the classifier-free guidance scale, $\gamma$. As discussed in Section 3, $\gamma$ balances the influence of the conditional and unconditional components of the denoiser. Since our goal is to generate solutions superior to those in the offline dataset, we hypothesized that a stronger guidance scale ($\gamma > 1$) would improve performance by forcing the model to adhere more closely to our novel conditioning points. Figure 3 plots the resulting HV-normalized by the performance of our default scale, $\gamma = 2.5$ across five different tasks.

The results in Figure 3 show that increasing the guidance scale yields surprisingly limited performance gains, with performance saturating or even slightly decreasing for $\gamma > 2.5$. We attribute this effect to two factors: 1) *The dataset is already reweighted* to prioritize high-performing samples. This means the diffusion model is inherently biased toward generating better solutions, even with weak conditioning. 2) *The conditioning points are near the data manifold.* Our reference-direction mechanism generates novel targets by extrapolating from the *best existing data*. While these targets are new, they remain close to the learned distribution. Consequently, the unconditional component of the model is already adept at generating similar high-quality solutions, reducing the marginal benefit of stronger conditional guidance.

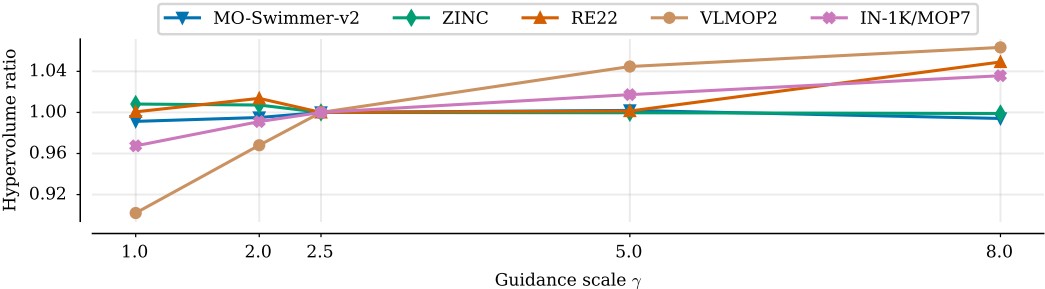

Figure 3: **Effect of the guidance scale $\gamma$ on the normalized HV ratio across five representative tasks.** Performance saturates quickly, showing limited benefit from strong guidance.

## 6 SUMMARY AND LIMITATIONS

This paper introduced Pareto-Conditioned Diffusion (PCD), a novel framework that reframes offline multi-objective optimization as a conditional sampling task. By learning to generate high-quality solutions conditioned directly on desired trade-offs, PCD elegantly sidesteps the need for explicit surrogate models or hand-crafted scalarization schemes. To make this approach effective, we introduced two key components: a multi-objective reweighting scheme to bias the model toward the Pareto front, and an NSGA-III-inspired mechanism to generate a diverse set of novel conditioning points from the offline data. Our extensive experiments demonstrated that PCD achieves competitive performance, driven by its remarkable consistency and robustness across a wide variety of benchmarks using a single set of hyperparameters.

Despite its strong performance, PCD has several limitations. On extremely high-dimensional continuous tasks such as MORL ($\sim$10,000 dimensions), performance is limited by applying an MLP denoiser directly to the parameter space. Future work could extend PCD by using a Latent Diffusion Model (Rombach et al., 2022) to operate in latent space or by adopting a Transformer-based denoiser (Peebles & Xie, 2023), both of which have demonstrated success in generating high-dimensional neural network parameters (Wang et al., 2024; Peebles et al., 2022). For discrete optimization tasks with categorical search spaces such as MONAS, the PCD framework can be readily extended using continuous-space diffusion methods for categorical data (Dieleman et al., 2022), which preserve the applicability of standard classifier-free guidance.

Alternatively, one could employ fully discrete diffusion models (Austin et al., 2021); while standard guidance is not applicable in this setting, recent work has developed effective extensions for discrete guidance (Nisonoff et al., 2025; Schiff et al., 2025). Finally, PCD can be naturally extended to combinatorial optimization problems (e.g., TSP, CVRP) by handling permutation constraints within the sampling loop (Sun & Yang, 2023). While outside the scope of this study, this could be achieved by combining constrained diffusion frameworks (Sun & Yang, 2023; Sanokowski et al., 2024) with the aforementioned discrete guidance techniques, effectively transferring our key contributions to the combinatorial setting.

## ACKNOWLEDGEMENTS

This work was supported by the Research Council of Finland (Decisions Nos. 353198, 353199, 357301, 339730, 362408, and 334600). Santeri Heiskanen was supported by the Ministry of Education and Culture's Doctoral Education Pilot in Finland (Decision No. VN/3137/2024-OKM-6; Finnish Doctoral Program Network in Artificial Intelligence, AI-DOC). We acknowledge the computational resources provided by the Aalto Science-IT project.

## 7 ETHICS STATEMENT

Our work, Pareto-Conditioned Diffusion (PCD), is a general-purpose optimization method intended for real-world applications in engineering and scientific design. However, we recognize its poten-

tial for misuse, as the method could be applied to harmful outcomes. We emphasize the need for oversight and safeguards to mitigate potential risks. Because PCD operates in an offline setting, it may also inherit or amplify biases from training data, underscoring the importance of careful dataset curation.

## 8 REPRODUCIBILITY STATEMENT

To ensure reproducibility, we provide supplementary materials with our submission, including:

- **Source Code:** A complete implementation of our proposed method, Pareto-Conditioned Diffusion (PCD).
- **Environment and Data:** A README file with instructions to set up the environment, along with links to all benchmark datasets and baseline implementations.
- **Scripts and Configurations:** Scripts to reproduce the results presented in the paper, including the exact training configurations, hyperparameters, and random seeds.

Our implementation is available at https://github.com/jatan12/PCD.

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

# A  APPENDIX

## A.1  RELATION BETWEEN DATASET QUALITY AND REWEIGHTING

As shown in our main ablation study (Section 5.4), our data reweighting scheme significantly improves performance on tasks like C10/MOP2 and MO-Hopper, yet is detrimental to performance on ZDT2. We hypothesized that this discrepancy stems from differences in the quality of the underlying offline datasets. To investigate this, we analyze the distribution of normalized dominance numbers for each task, shown in Figure 4. A narrow distribution suggests that most samples are of similar, relatively high quality. In contrast, a wide, right-skewed distribution indicates a large variance in sample quality, with a long tail of low-performing data points.

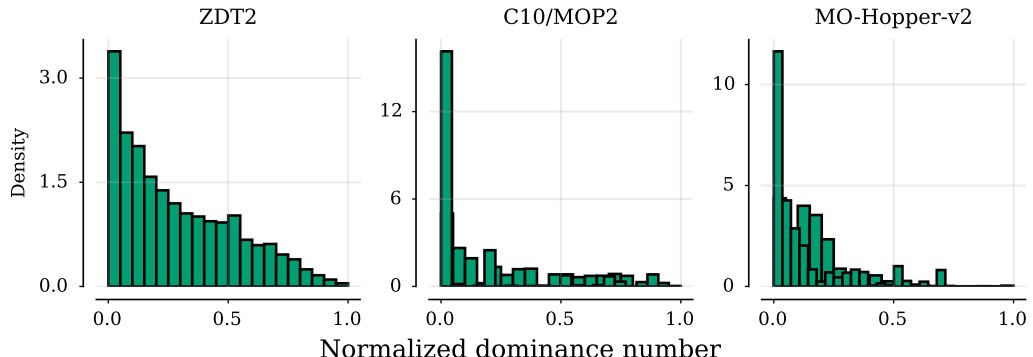

Figure 4: **Normalized dominance number distributions reveal dataset quality differences.** The narrow distributions of C10/MOP2 and MO-Hopper indicate consistently high-quality datasets, whereas the long-tailed distribution of ZDT2 signals high variance in sample quality.

Figure 4 clearly supports our hypothesis. The C10/MOP2 and MO-Hopper tasks exhibit narrow distributions, concentrated near zero. ZDT2, however, shows a much wider, long-tailed distribution, confirming it contains a more varied set of samples. While a dataset with high variance in quality like ZDT2 may seem like an ideal candidate for reweighting, our results show the opposite can be true. When the sample quality is highly varied, our reweighting scheme assigns a tiny, near-zero weight to the vast majority of points in the long tail. This can effectively remove a significant portion of the training data, which, despite being lower-quality, may still contain valuable learning signal about the data manifold. This raises a natural follow-up question: could this issue be mitigated by tuning the reweighting hyperparameters?

## A.2  REWEIGHING SENSITIVITY ANALYSIS

Based on our analysis in Section A.1, we hypothesized that the performance degradation on high-variance datasets like ZDT2 could be mitigated by adjusting the reweighting hyperparameters. Specifically, increasing the temperature parameter $\tau$ in Equation 6 should make the weighting distribution more uniform, effectively "softening" the reweighting and preserving more of the learning signal from lower-quality samples. To validate this, we perform a sensitivity analysis on $\tau$ for the same three tasks: ZDT2, C10/MOP2, and MO-Hopper.

Figure 5 plots the normalized hypervolume as a function of $\tau$. The results are normalized by the performance of our default value, $\tau = 0.05$. As predicted, the effect of $\tau$ is highly task-dependent. For ZDT2, the task with the highest sample variance, increasing the temperature yields a substantial performance improvement of up to 15%. In contrast, for the low-variance C10/MOP2 dataset, performance is almost entirely insensitive to $\tau$. The MO-Hopper task, which has a moderate variance, shows a modest improvement.

This outcome confirms our intuition: *the higher the variance in dataset quality, the higher the temperature $\tau$ is required to achieve optimal performance.* Although we explored systematic ways to tune $\tau$ (e.g., estimating it from the standard deviation or CDF of the dominance number distribution), we

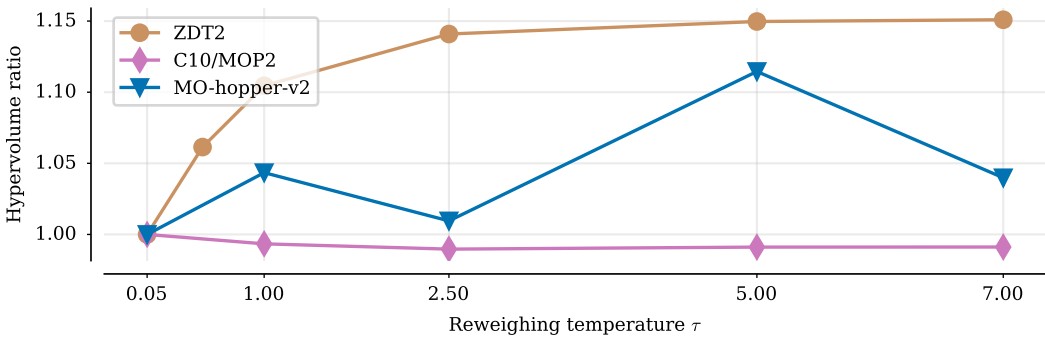

Figure 5: **Effect of the reweighting temperature $\tau$ on normalized performance.** For the high-variance ZDT2 dataset, increasing $\tau$ significantly improves performance. The effect is minimal on the low-variance datasets.

found the default value ($\tau = 0.05$) to be robust across the majority of tasks. However, for outlier datasets with high variance like ZDT2, we recommend a simple visual heuristic: *if the distribution of dominance numbers exhibits a "long tail," increasing $\tau$ effectively "softens" the reweighting to preserve more training data.* We note that the other reweighting hyperparameter, $K$, which controls smoothing over bin sizes, was observed to have a negligible impact on performance in preliminary experiments, and thus was not included in this analysis.

### A.3 TRAINING DETAILS

We build upon the open-source EDM implementation (Karras et al., 2022) introduced in SynthER (Lu et al., 2023), adapting it to our specific training setup and dataset. Following SynthER, our denoising network adopts the MLP architecture with skip connections from the previous layer as proposed in Tolstikhin et al. (2021), with the diffusion noise level encoded using a Random Fourier Feature (RFF) embedding (Rahimi & Recht, 2007). The network has a base width 512 and depth of 4, and is trained with batch size of 512 for up to 12k steps with early stopping.

We apply CFG dropout (Ho & Salimans, 2022) during training with probability 0.25 and use the same denoiser configuration for all tasks. We maintain an exponential moving average (EMA) of the model parameters (Karras et al., 2022) and use the EMA weights for sampling to improve stability (Karras et al., 2024). Sampling is performed with the EDM stochastic differential equation (SDE) sampler using default hyperparameters, increasing the number of diffusion timesteps to 1024 to enhance sample quality. The default hyperparameters for all components of our PCD framework—including the denoiser network, the EDM sampler, our multi-objective reweighting scheme, and our conditioning point generation mechanism—are summarized in Table 3.

Table 3: Default hyperparameters for PCD framework including Residual MLP Denoiser for reweighted training (left) and EDM Sampling (right). If multiple values were used, **bolded** value indicates default value.

| Residual MLP Denoiser | | EDM | |
|---|---|---|---|
| **Parameter** | **Value(s)** | **Parameter** | **Value(s)** |
| No. of Bins | 30 | No. of Diffusion Steps | 1024 |
| $K$ | 10 | $\sigma_{min}$ | 0.002 |
| $\tau$ | 0.05 | $\sigma_{max}$ | 80 |
| No. of Layers | 4 | $S_{churn}$ | 80 |
| Width | 512 | $S_{tmin}$ | 0.05 |
| Batch Size | 512 | $S_{tmax}$ | 50 |
| RFF Dim | 16 | $S_{noise}$ | 1.003 |
| Activation | ReLU | Query Budget $Q$ | 256 |
| Optimizer | AdamW | No. of Reference Directions $L$ | 32 |
| Learning Rate | $3 \times 10^{-4}$ | CFG Dropout Prob. | 0.25 |
| Learning Rate Schedule | Cosine Annealing | CFG Guidance Scale $\gamma$ | $\{1.0, 2.0, \mathbf{2.5}, 5.0, 8.0\}$ |
| Denoiser Training Steps | 12,000 (early stopping) | | |

A.4 COMPUTATIONAL COST

A key component of our reweighting strategy is the computation of the dominance number $o(\boldsymbol{x})$ (Eq. 5) for each sample in the offline dataset. Naively, this calculation scales quadratically with the dataset size, $\mathcal{O}(N^2 m)$. However, we emphasize that this is a one-time preprocessing step performed prior to training. For standard offline MOO benchmarks (up to $N = 60,000$), this calculation is inexpensive in practice when utilizing optimized non-dominated sorting algorithms (e.g., from the `pygmo` library (Biscani & Izzo, 2020)). Empirically, computing the dominance numbers for our largest dataset (RE34) takes less than 20 seconds, as shown in Table 4. While the quadratic cost did not pose issues for the benchmarks considered in this work, it may become a limitation for extremely large datasets, since the worst-case complexity remains quadratic.

To examine inference costs, we report the wall-clock time for sampling, averaged over three seeds, across four representative tasks for PCD and the predictor-guided ParetoFlow baseline (Yuan et al., 2025). All experiments for this study were conducted on a workstation with an Intel i5-12600K CPU, 64GB of memory, and an NVIDIA GeForce RTX3080 GPU. Table 4 highlights the key finding: PCD sampling generates a batch of 256 solutions faster than ParetoFlow on most tasks. Although PCD employs CFG (Ho & Salimans, 2022), which requires two forward passes per step, it avoids the computationally expensive and memory-intensive backpropagation through the network required by the baseline's predictor-guidance (Dhariwal & Nichol, 2021).

Table 4: **Computational cost comparison.** We report the one-time dataset reweighting cost (seconds) and sampling times (seconds) averaged over 3 runs. PCD demonstrates faster sampling speeds compared to the generative baseline ParetoFlow across four representative tasks, while reweighting adds negligible overhead.

| Method | RE34 | MO-Hopper-v2 | Molecule | C10/MOP2 |
|---|---|---|---|---|
| | **Dataset Info** | | | |
| No. of Samples | 60,000 | 4,500 | 49,001 | 26,316 |
| No. of Objectives | 3 | 2 | 2 | 3 |
| No. of Dimensions | 5 | 10,184 | 32 | 31 |
| Reweighting Time (sec) | 17.3±0.147 | 0.090±0.000 | 11.8±0.019 | 2.94±0.0125 |
| | **Sampling Time (sec)** | | | |
| PCD | **1.96±0.13** | **3.22±0.01** | **1.80±0.02** | **2.02±0.14** |
| ParetoFlow | 6.08±0.06 | 17.20±0.61 | 36.91±1.99 | 5.81±0.04 |

A.5 DETERMINISTIC SAMPLER VS. STOCHASTIC SAMPLER

Recent work has established a strong theoretical connection between diffusion models (Ho et al., 2020; Song et al., 2021b) and flow matching (Lipman et al., 2023; Liu et al., 2023), showing that they can be unified under the framework of continuous-time generative models (Gao et al., 2025; Albergo et al., 2025). In particular, when using Gaussian probability flow paths, the sampling process for both frameworks is governed by an ODE that maps a Gaussian noise distribution to the data distribution. Consequently, using a diffusion model with a deterministic ODE sampler is mathematically equivalent to the sampling process of standard Gaussian flow matching (Gao et al., 2025). The primary practical differences lie in the parameterization of the vector field, the noise schedule, and the weighting of the training objective (Gao et al., 2025). These design choices have been extensively studied within the diffusion context, most notably in the EDM paper (Karras et al., 2022). This work uses the parameterization derived in EDM.

To validate our choice of sampler, we compare our default stochastic EDM (an SDE-like sampler) against its deterministic counterpart (ODE). The results in Table 5 demonstrate that the stochastic sampler outperforms the deterministic one on most tasks. We hypothesize that the randomness inherent to the stochastic EDM solver aids exploration, which is particularly beneficial for discovering diverse, high-quality solutions in offline MOO. The notable exception is the discrete Regex task, where the deterministic sampler performs better. We hypothesize that this is due to the final

decoding step: mapping continuous logits to categorical tokens is sensitive to noise, and the deterministic trajectory of the ODE sampler leads to more stable token sequences (Song et al., 2021a). Consequently, we adopt the stochastic sampler as the default configuration, as Regex is the only exception.

Table 5: Average 100th percentile HV ($\uparrow$) on 5 different tasks comparing EDM with stochastic and deterministic samplers. **Bold** and underlined rows indicate the best and runner up methods respectively. The default stochastic sampler shows superior performance on most tasks.

| Method | C10/MOP2 | MO-Hopper-v2 | RE34 | Regex | ZDT2 |
|---|---|---|---|---|---|
| Stochastic PCD | **10.59±0.04** | 5.95±0.18 | **10.17±0.04** | 4.80±0.87 | 6.25±0.06 |
| Deterministic PCD | 10.54±0.03 | 4.47±0.00 | 9.47±0.28 | 5.56±0.48 | 5.00±0.04 |
| Stochastic PCD (w/o reweighting) | 10.47±0.01 | **6.01±0.58** | 10.11±0.03 | 5.55±0.25 | **7.89±0.19** |
| Deterministic PCD (w/o reweighting) | 10.49±0.06 | 4.47±0.00 | 9.24±0.26 | **5.68±0.31** | 4.98±0.03 |

### A.6 FURTHER ABLATIONS ON SAMPLING HYPERPARAMETERS

To further analyze the components of the proposed method, we perform multiple ablations on the key design choices related to generating conditioning points. Specifically, we examine the effect of: 1) varying the number of selected base points (Table 6); 2) increasing the noise scale and extrapolation distance applied to the generated conditioning points (Figure 6); and 3) changing the method used for generating direction vectors (Table 7).

#### A.6.1 NUMBER OF CONDITIONING POINTS

We begin by examining the number of conditioning points selected at the second stage of conditioning point generation. Recall that if the number of selected points $J$ is less than the number of generated solutions $Q$ (as is often the case), multiple conditioning points will be generated from the same selected point. Therefore, the more points are selected, the fewer new points are generated for each selected point. As seen from Table 6, increasing the number of conditioning points tends to improve performance, with the exception of Regex, where using $J = 4$ points performs the best. This indicates that choosing a sufficient number of points from the existing dataset is required for achieving reasonable performance.

Table 6: Average 100th percentile Hypervolume ($\uparrow$) on 5 different tasks using varying numbers of conditioning points $J$. **Bold** and underlined rows indicate the best and runner up methods respectively. Increasing the number of selected points generally yields higher performance, with $J = 32$ achieving the best results on the majority of tasks.

| No. cond. points | C10/MOP2 | MO-Hopper-v2 | RE34 | Regex | ZDT2 |
|---|---|---|---|---|---|
| $J$=4 | 10.31±0.12 | 4.47±0.00 | 6.56±0.83 | **5.26±0.70** | 4.87±0.08 |
| $J$=8 | 10.37±0.08 | 4.47±0.00 | 7.19±0.74 | 4.69±0.75 | 4.92±0.05 |
| $J$=16 | 10.41±0.01 | 4.47±0.00 | 8.33±0.59 | 4.69±0.75 | 4.97±0.03 |
| $J$=32 | **10.59±0.04** | **5.95±0.18** | 10.17±0.04 | 4.80±0.87 | **6.25±0.06** |
| $J$=64 | 10.42±0.01 | 4.47±0.00 | **11.02±0.26** | 4.81±0.72 | 4.98±0.04 |

#### A.6.2 NOISE SCALE AND EXTRAPOLATION DISTANCE

Figure 6 illustrates the effect of increasing the noise scale and extrapolation distance, the two primary parameters controlling the generation of novel conditioning points. Intuitively, increasing the noise scale produces a more diverse set of conditioning points, while increasing the extrapolation distance pushes the targets further from the original data distribution. As shown in Figure 6, increasing the noise scale has a limited yet positive impact on performance without significant downsides, likely due to the increased diversity of the generated targets. Conversely, increasing the extrapolation distance offers negligible gains and significantly increases run-to-run variance on MO-Hopper. As discussed in Section 5.3, MORL tasks proved challenging for current approaches; consequently, pushing conditioning targets further from the training distribution increases model uncertainty. For

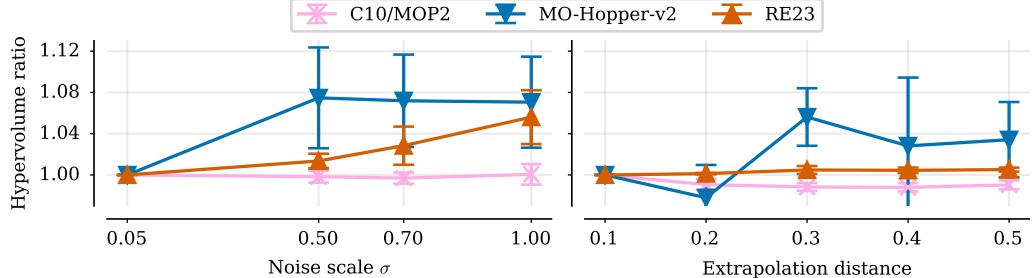

Figure 6: **Impact of noise scale and extrapolation distance on conditioning point generation.**
**Left**: Increasing the noise scale has a limited, yet positive impact on performance. **Right**: Increasing
the extrapolation distance leads to diminishing returns.

the remaining tasks, we hypothesize that the limited gains stem from the model's inability to gener-
alize effectively to regions farther away from the training distribution.

### A.6.3 RIESZ s-ENERGY VS. DAS-DENNIS

Generating conditioning points requires a set of reference direction vectors. Table 7 compares our
default method, Riesz s-Energy (Hardin & Saff, 2005), against the classical Das-Dennis method
(Das & Dennis, 1998). While the Riesz s-Energy method allows one to precisely set the number
of generated direction vectors, it requires an optimization procedure. In contrast, the Das-Dennis
method is simpler and requires no optimization. However, it does not allow for explicit control over
the number of generated direction vectors. As shown in Table 7, while Das-Dennis slightly outper-
forms Riesz s-Energy on most tasks, the performance difference is marginal. This demonstrates that
PCD is robust to the underlying strategy used for generating direction vectors.

Table 7: Average 100th percentile HV ($\uparrow$) comparing Riesz s-Energy (default) and Das-Dennis
methods for generating direction vectors. **Bold** rows indicate the best method. The simpler Das-
Dennis method yields slightly better results, though the difference is minimal, indicating robustness.

| Method | C10/MOP2 | MO-Hopper-v2 | RE34 | Regex | ZDT2 |
|---|---|---|---|---|---|
| Riesz s-Energy | **10.59±0.04** | 5.95±0.18 | 10.17±0.04 | 4.80±0.87 | 6.25±0.06 |
| Das-Dennis | 10.58±0.04 | **6.16±0.30** | **10.23±0.12** | **5.21±0.67** | **6.32±0.13** |

### A.6.4 NUMBER OF DENOISING STEPS

Finally, we perform an ablation to study the effect of increasing the number of denoising steps.
While increasing the sampling budget typically improves generation quality in standard diffusion
literature (Nichol & Dhariwal, 2021; Salimans & Ho, 2022), Figure 7 shows that this is not the
case here. With the exception of Regex, the proposed method achieves consistent performance
regardless of the number of denoising steps. We attribute this to the fact that we are generating points
relatively close to the training distribution, allowing the model to produce high-quality samples even
with fewer steps. The deviation in Regex aligns with our discussion in Section A.5: discrete data
requires a decoding step, which is sensitive to approximation errors. Although the MONAS task
(C10/MOP2) is also discrete, it is far simpler, as each of its decision variables takes only 3 values.
In contrast, Regex variables have *up to 72 options*, making the decoding process far more susceptible
to the small errors caused by fewer sampling steps.

### A.7 CASE STUDY 1: THE EFFECTIVENESS OF PARETO-CONDITIONING

To provide a qualitative understanding of our Pareto-conditioning mechanism, we present a case
study on two contrasting tasks: MO-Hopper, a challenging benchmark where our method struggles,

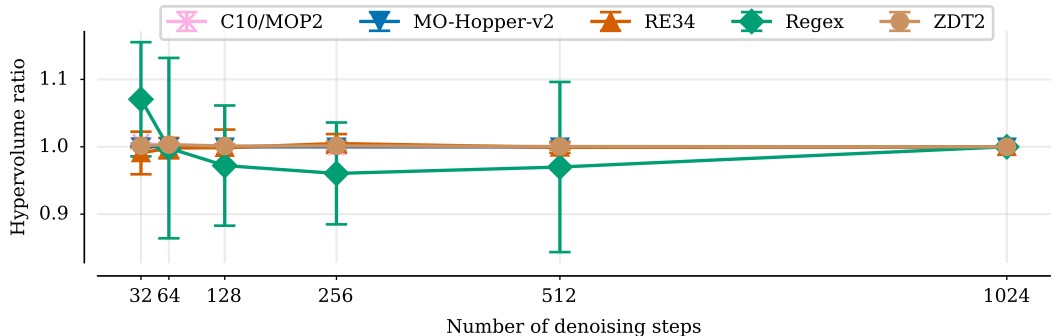

Figure 7: **Hypervolume ratio as a function of denoising steps.** Performance remains consistent across increasing numbers of steps for most tasks, demonstrating that PCD generates high-quality solutions efficiently even with fewer sampling steps.

and RE21, a task where PCD significantly outperforms all baselines. These cases correspond to the quantitative results presented in Table 16 and Table 12, respectively.

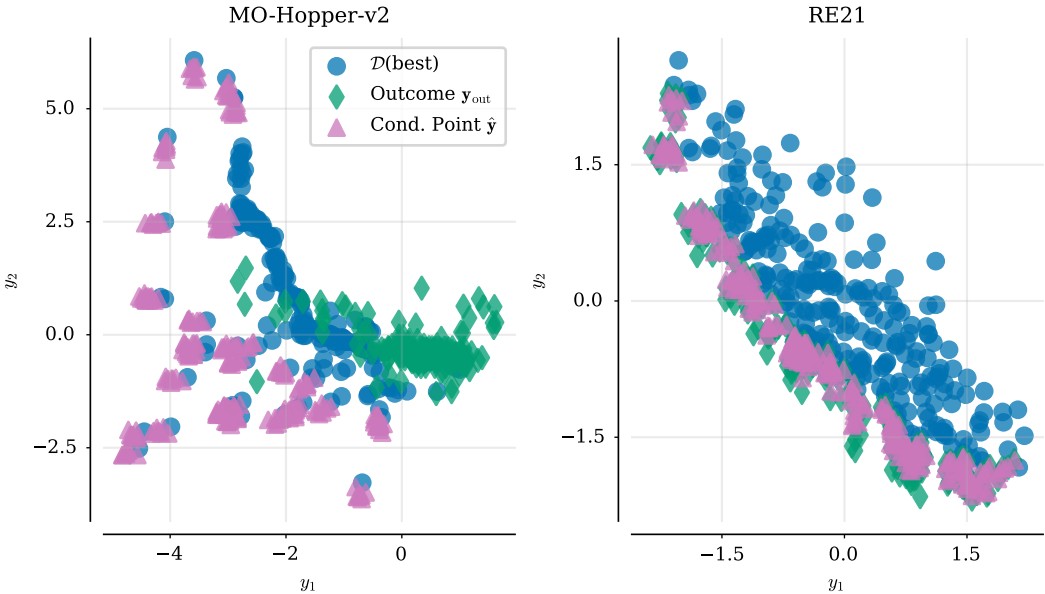

Figure 8: **Qualitative analysis of Pareto-conditioning effectiveness. Left** (MO-Hopper): On this high-dimensional task, the model struggles to generate solutions that improve upon the first objective. **Right** (RE21): On this lower-dimensional task, the conditioning is highly effective: the generated points ($\boldsymbol{y}_{\text{out}}$) closely align with and even outperform their conditioning targets ($\hat{\boldsymbol{y}}$).

Figure 8 visualizes the relationship between the best solutions from the original dataset $\mathcal{D}(\text{best})$, our generated conditioning points $\hat{\boldsymbol{y}}$, and the objective values of the final generated solutions $\boldsymbol{y}_{\text{out}}$. On MO-Hopper, the model's struggle is evident. Despite being conditioned on targets $\hat{\boldsymbol{y}}$ with better performance on the first objective $y_1$, the generated points $\boldsymbol{y}_{\text{out}}$ fail to achieve significant improvements, likely due to the difficulty of generalizing in the task's high-dimensional search space. In stark contrast, the conditioning mechanism is highly effective on RE21. Here, the generated points $\boldsymbol{y}_{\text{out}}$ consistently match or even Pareto-dominate their respective conditioning targets $\hat{\boldsymbol{y}}$. This demonstrates that when the underlying data distribution is well-captured by the model, PCD can successfully generalize to superior regions of the Pareto front.

To further study the capabilities of PCD, we examine its ability to generate points *within the training distribution*. More specifically, we partition the training dataset into fronts $(F_1, \ldots, F_M)$ using non-

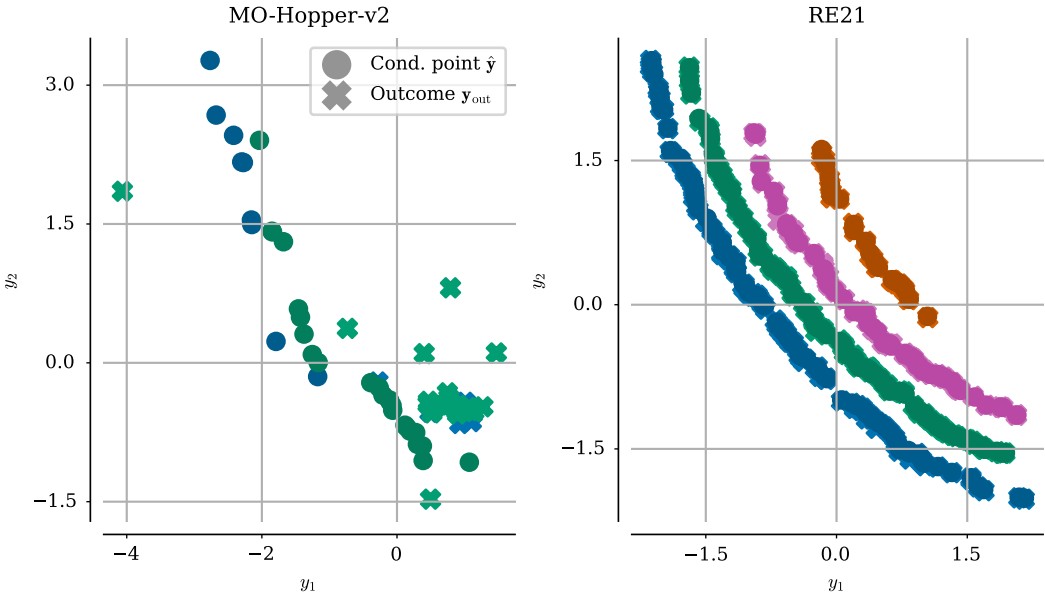

Figure 9: **Analysis of the model's ability to reconstruct the original data distribution.** Colors represent distinct non-dominated fronts ($F_i$) extracted from the training set. The model is conditioned on the objective vectors of the original data (circles, $\hat{y}$); successful reconstruction is achieved if the generated solutions (crosses, $y_{\text{out}}$) overlap with these circles. **Left** (MO-Hopper): In this high-dimensional task, the model is unable to capture the original data distribution it was trained on. **Right** (RE21): In this simpler task, the model faithfully reconstructs the original conditioning points.

dominated sorting. Then, we condition the model on the objective vectors from these fronts $F_i$ to validate whether it can reconstruct the data distribution it was trained on. Figure 9 showcases a sharp contrast: while in the simpler, lower-dimensional RE21 task, PCD *captures the underlying distribution almost perfectly*, it fails to do so in the more complex, high-dimensional MO-Hopper task. We highlight that while the positive results on RE21 demonstrate the potential of the proposed approach, the MO-Hopper results indicate a clear need for further development to enable generalization to very high-dimensional data distributions.

## A.8 CASE STUDY 2: SCALABILITY TO MANY-OBJECTIVE PROBLEMS

While our main experiments demonstrate PCD's strong performance on standard tasks, most of these benchmarks are limited to 2 or 3 objectives. To investigate PCD's scalability to many-objective problems, we conduct an ablation study on the synthetic DTLZ1 and DTLZ7 functions, varying the number of objectives $m \in \{3, 4, 5, 6\}$. To facilitate this experiment, we collect offline datasets following the protocol described by Xue et al. (2024). Specifically, we used the NSGA-III (Deb & Jain, 2014) algorithm to collect the dataset of $N = 60,000$ samples per task, aggregated across multiple random seeds.

Table 8 reports the relative improvement in Hypervolume (HV) achieved by PCD over the best solution set available in the offline dataset ($\mathcal{D}(\text{best})$). By analyzing the relative improvement rather than raw HV values, we normalize for dataset quality and objective scaling, allowing us to isolate PCD's performance trend as dimensionality increases. As shown in Table 8, PCD achieves consistent improvements over the baseline regardless of the number of objectives. This highlights the scalability of our reweighting and conditioning mechanisms, demonstrating that PCD effectively generalizes even in high-dimensional objective spaces.

Table 8: **Scalability analysis on many-objective DTLZ tasks.** We report the relative Hypervolume improvement (↑) of PCD over the best offline dataset solutions ($\mathcal{D}$(best)). PCD achieves consistent performance improvements regardless of the number of objectives ($m$).

| Task | No. objectives | | | |
|---|---|---|---|---|
| | 3 | 4 | 5 | 6 |
| DTLZ1 | 1.29±0.07 | 1.25±0.04 | 1.84±0.16 | 1.37±0.08 |
| DTLZ7 | 1.33±0.02 | 1.17±0.01 | 1.19±0.02 | 1.21±0.04 |

## A.9    DETAILED LOOK AT CONDITIONING POINT GENERATION

Section 4 introduced the procedure to generate a set of diverse conditioning points $\hat{\boldsymbol{y}}$ by utilizing a procedure similar to the survival selection from NSGA-III (Deb & Jain, 2014), which is detailed in Algorithm 2. We begin by generating $L$ direction vectors via Riesz s-Energy method (Hardin & Saff, 2005), which partition the objective space $\mathbb{R}^m$. Next, our goal is to select a diverse subset of well-performing points from the objective space and assign these points to one of the direction vectors. To do so, an empty set $S$ is initialized to hold pairs of points and direction vectors ($\boldsymbol{y}_i, \boldsymbol{w}_i$). Then, a set of fronts ($F_1, \ldots, F_M$) is obtained via non-dominated sorting. The procedure continues by iterating over each front $F_i$. During each iteration, we identify the point with the shortest perpendicular distance to each direction vector and assign it to that vector (lines 8-13 in Alg. 2). Finally, the remaining points in $F_i$ (which have no direction vector assigned to them), are iterated over one-by-one and assigned to the direction vector that currently has the fewest assigned points (lines 14-22 in Alg. 2).

After obtaining the set of (point, direction vector) pairs $S$, the points are extrapolated towards the corresponding direction vectors. Finally, Gaussian noise is added to the points to ensure further diversity. The procedure described in Algorithm 2 follows very closely to the survival procedure of NSGA-III with one key difference: in NSGA-III the points in the current population are assigned to the closest direction vector, and only if there is a need for more points, the points get assigned to the direction vectors with the least number of points. On the contrary, in our procedure, only $L$ points from a front get automatically assigned the direction vector closest to them. The rest of the points in the current front are assigned based on the number of points each direction vector has currently. This is crucial for ensuring that the conditioning points have an adequate coverage of the objective space in cases where the offline dataset itself has a poor coverage of the objective space.

## A.10    USE OF LARGE LANGUAGE MODELS

Finally, during the preparation of this manuscript, large language models (LLMs) were used to assist in writing and literature discovery, but all content was reviewed and verified by the authors, who are solely responsible for the scientific claims of this work.

## A.11    100TH PERCENTILE RESULTS

Tables 9, 10, 11, 12, 13, 14, 15 and 16 showcase the per task 100th percentile hypervolume (↑) across all considered tasks. In each task, the best performing model is **bolded**, while the runner up is underlined.

## A.12    75TH PERCENTILE RESULTS

Tables 17, 18, 19, 20, 21, 22, 23 and 24 showcase the per task 75th percentile hypervolume (↑) across all considered tasks. In each task, the best performing model is **bolded**, while the runner up is underlined.

---

**Algorithm 2** Generating Diverse Conditioning Points

    **Input:** Dataset objectives $Y = \{y_i\}_{i=1}^N$, Number of reference vectors $L$, Query budget $Q$
    **Output:** Set of $Q$ conditioning points $\hat{Y}$
  1: $W \leftarrow$ GenerateReferenceVectors($L$)          ▷ e.g., via Riesz s-Energy method
  2: $(F_1, \ldots, F_M) \leftarrow$ NonDominatedSort($Y$)          ▷ Rank data points into fronts
  3: $S \leftarrow \emptyset$          ▷ Initialize the set of selected (point, vector) pairs
  4: $\rho \leftarrow$ Initialize niche counts for each $w \in W$ to zero
  5: $i \leftarrow 1$
  6: **while** $|S| < Q$ **do**
  7:      $P_i \leftarrow \emptyset$          ▷ Initialize points to be considered from current front
  8:      **for** each reference vector $w \in W$ **do**
  9:          Find point $y \in F_i$ with minimum perpendicular distance to $w$
 10:          Add $(y, w)$ to $S$
 11:          Add $y$ to temporary set $P_i$
 12:          $\rho[w] \leftarrow \rho[w] + 1$          ▷ Update niche counts for each reference vector
 13:      **end for**
 14:      **for** $y \in F_i \setminus P_i$ **do**
 15:          Find reference vector $w_{\min}$ with the minimum niche count $\rho[w_{\min}]$
 16:          $S \leftarrow S \cup \{(y^*, w_{\min})\}$          ▷ Add the selected pair to our main set
 17:          $\rho[w_{\min}] \leftarrow \rho[w_{\min}] + 1$          ▷ Increment the niche count
 18:          $P_t \leftarrow P_t \setminus \{y^*\}$          ▷ Remove the point from consideration
 19:          **if** $|S| = Q$ **then**
 20:              **break**          ▷ Exit if we have collected enough points
 21:          **end if**
 22:      **end for**
 23:      $i \leftarrow i + 1$          ▷ Move to the next front
 24: **end while**
 25: $Y_{\text{extrapolated}} \leftarrow$ ExtrapolatePoints($S$)          ▷ Extrapolate each point along its vector
 26: $\hat{Y} \leftarrow Y_{\text{extrapolated}} + \mathcal{N}(0, \sigma_{\text{noise}}^2 \mathbb{I})$          ▷ Add noise for diversity
 27: **return** $\hat{Y}$

---

Table 9: Average 100th percentile hypervolume for synthetic tasks (part 1)

| Method | ZDT1 | ZDT2 | ZDT3 | ZDT4 | ZDT6 |
|---|---|---|---|---|---|
| $\mathcal{D}$(best) | 4.17 | 4.68 | 5.15 | **5.46** | **4.61** |
| E2E + GN | 2.81±0.00 | 3.17±0.00 | 3.75±0.00 | 4.87±0.05 | 3.82±0.03 |
| E2E + PC | 2.81±0.00 | 3.17±0.00 | 3.75±0.00 | 4.84±0.09 | 3.78±0.08 |
| E2E | 2.81±0.00 | 3.17±0.00 | 3.76±0.00 | 4.82±0.02 | 3.83±0.01 |
| MOBO | 2.38±0.01 | 2.76±0.00 | 3.13±0.05 | 4.93±0.09 | 3.76±0.00 |
| MH + GN | 2.81±0.00 | 3.17±0.00 | 3.76±0.00 | 4.83±0.05 | 3.74±0.04 |
| MH + PC | 2.41±0.27 | 2.68±0.01 | 3.68±0.16 | 4.83±0.04 | 3.76±0.04 |
| MH | 2.81±0.00 | 3.17±0.00 | 3.76±0.00 | 4.87±0.02 | 3.83±0.01 |
| MM + COMs | 2.81±0.00 | 3.17±0.00 | 3.75±0.00 | 4.84±0.03 | 3.82±0.04 |
| MM + ICT | 2.81±0.00 | 3.17±0.00 | 3.76±0.00 | 4.86±0.04 | 3.83±0.02 |
| MM + IOM | 2.81±0.00 | 3.17±0.00 | 3.73±0.05 | 4.85±0.03 | 3.84±0.01 |
| MM + TM | 2.81±0.00 | 3.17±0.00 | 3.73±0.05 | 4.80±0.10 | 3.84±0.00 |
| MM | 2.81±0.00 | 3.17±0.00 | 3.76±0.00 | 4.85±0.06 | 3.82±0.00 |
| ParetoFlow | 4.15±0.04 | 5.71±0.21 | 5.22±0.11 | 4.86±0.08 | 4.51±0.05 |
| PCD (ours) | **4.47±0.02** | **6.25±0.06** | **5.36±0.36** | 5.23±0.06 | 4.56±0.05 |

Table 10: Average 100th percentile hypervolume for synthetic tasks (part 2)

| Method | DTLZ1 | DTLZ7 | OMNITEST | VLMOP1 | VLMOP2 | VLMOP3 |
|---|---|---|---|---|---|---|
| $\mathcal{D}$(best) | 10.60 | 8.56 | 4.53 | **0.41** | 1.78 | 45.65 |
| E2E + GN | 10.63±0.00 | 8.95±0.00 | 4.55±0.14 | 0.29±0.00 | 2.80±0.16 | 41.93±1.13 |
| E2E + PC | 10.63±0.00 | 8.95±0.00 | 4.69±0.06 | 0.29±0.00 | 3.34±0.01 | 42.45±0.00 |
| E2E | 10.61±0.01 | 8.95±0.00 | 4.69±0.05 | 0.29±0.00 | 3.36±0.00 | 42.45±0.00 |
| MOBO | 10.40±0.06 | 3.96±0.10 | 4.76±0.01 | 0.29±0.00 | 2.08±0.16 | 38.99±0.14 |
| MH + GN | 10.63±0.01 | 8.95±0.00 | 4.61±0.09 | 0.08±0.03 | 3.15±0.07 | 30.14±6.91 |
| MH + PC | 10.63±0.00 | 8.95±0.00 | 4.48±0.22 | 0.20±0.04 | 2.98±0.09 | 41.85±0.91 |
| MH | 10.63±0.00 | 8.95±0.00 | 4.61±0.09 | 0.29±0.00 | 3.32±0.01 | 42.45±0.00 |
| MM + COMs | 10.62±0.00 | 8.95±0.00 | 4.54±0.15 | 0.23±0.13 | 1.68±0.04 | 42.44±0.02 |
| MM + ICT | 10.63±0.00 | 8.95±0.00 | 4.56±0.08 | 0.29±0.00 | 3.35±0.01 | 42.43±0.01 |
| MM + IOM | 10.63±0.00 | 8.95±0.00 | 4.72±0.03 | 0.29±0.00 | 3.32±0.03 | 42.45±0.01 |
| MM + TM | 10.63±0.00 | 8.95±0.01 | 4.59±0.10 | 0.29±0.00 | 3.35±0.00 | 42.41±0.01 |
| MM | 10.63±0.00 | 8.95±0.00 | 4.71±0.06 | 0.29±0.00 | 3.35±0.01 | 42.45±0.00 |
| ParetoFlow | 11.15±0.44 | 10.54±0.44 | **4.78±0.00** | N/A | **4.22±0.02** | **45.92±0.00** |
| PCD (ours) | **13.63±0.78** | **11.35±0.18** | 4.76±0.01 | 0.09±0.01 | 2.93±0.28 | 45.91±0.01 |

Table 11: Average 100th percentile hypervolume for scientific design tasks

| Method | Molecule | RFP | Regex | Zinc |
|---|---|---|---|---|
| $\mathcal{D}$(best) | 2.91±0.00 | 7.11±0.00 | 3.96±0.00 | 4.52±0.00 |
| E2E + GN | 2.77±0.07 | 7.88±0.61 | 4.30±0.35 | 4.65±0.01 |
| E2E + PC | 2.91±0.00 | 7.22±0.02 | 4.74±0.20 | **4.73±0.04** |
| E2E | 2.91±0.00 | **8.11±0.49** | 3.98±0.00 | 4.71±0.07 |
| MOBO | 1.58±0.02 | 7.21±0.00 | **6.76±0.28** | 4.60±0.00 |
| MH + GN | 2.91±0.00 | 8.10±0.50 | 4.19±0.20 | 4.62±0.06 |
| MH + PC | 2.91±0.00 | 7.41±0.49 | 4.52±0.45 | 4.65±0.03 |
| MH | 2.91±0.00 | 7.21±0.02 | 3.98±0.00 | 4.66±0.00 |
| MM + COMs | 2.91±0.00 | 7.66±0.61 | 4.61±0.30 | 4.72±0.05 |
| MM + ICT | 2.91±0.00 | 7.87±0.61 | 4.30±0.35 | 4.65±0.01 |
| MM + IOM | 2.91±0.00 | 7.66±0.62 | 4.83±0.00 | 4.69±0.05 |
| MM + TM | 2.87±0.06 | 7.83±0.65 | 4.83±0.00 | 4.64±0.01 |
| MM | 2.91±0.00 | 7.21±0.03 | 3.87±0.24 | 4.66±0.00 |
| ParetoFlow | 2.06±0.31 | 7.07±0.09 | 5.11±0.39 | 4.40±0.03 |
| PCD (ours) | **4.07±0.51** | 7.24±0.03 | 4.80±0.87 | 4.70±0.04 |

Table 12: Average 100th percentile hypervolume for RE tasks (part 1)

| Method | RE21 | RE22 | RE23 | RE24 | RE25 | RE31 | RE32 |
|---|---|---|---|---|---|---|---|
| $\mathcal{D}$(best) | 4.10 | 4.78 | 4.75 | 4.60 | 4.79 | 10.60 | 10.56 |
| E2E + GN | 2.60±0.00 | 3.18±0.00 | 0.00±0.00 | 1.66±0.00 | 3.81±0.17 | 5.06±0.73 | 4.96±0.13 |
| E2E + PC | 2.60±0.00 | 1.91±1.74 | 0.00±0.00 | 1.66±0.00 | 3.29±0.71 | 6.54±0.13 | 7.07±0.32 |
| E2E | 2.60±0.00 | 3.18±0.00 | 0.00±0.00 | 1.66±0.00 | 3.93±0.07 | 6.62±0.00 | 7.14±0.20 |
| MOBO | 1.98±0.02 | 0.00±0.00 | 0.00±0.00 | 0.00±0.00 | 0.00±0.00 | 0.00±0.00 | 0.00±0.00 |
| MH + GN | 2.58±0.03 | 1.91±1.74 | 0.00±0.00 | 0.00±0.00 | 3.19±0.61 | 5.97±0.36 | 4.37±0.73 |
| MH + PC | 2.60±0.00 | 1.91±1.74 | 0.00±0.00 | 0.00±0.00 | 0.00±0.00 | 6.42±0.46 | 5.54±3.16 |
| MH | 2.60±0.00 | 3.18±0.00 | 0.00±0.00 | 1.66±0.00 | 3.41±0.73 | 6.62±0.00 | 7.28±0.00 |
| MM + COMs | 2.59±0.00 | 1.64±1.60 | 0.00±0.00 | 1.66±0.00 | 3.98±0.00 | 5.67±1.56 | 7.28±0.00 |
| MM + ICT | 2.60±0.00 | 3.18±0.00 | 0.00±0.00 | 1.66±0.00 | 3.98±0.00 | 6.54±0.08 | 7.28±0.00 |
| MM + IOM | 2.60±0.00 | 3.18±0.00 | 0.00±0.00 | 1.66±0.00 | 3.93±0.07 | 6.62±0.00 | 7.28±0.00 |
| MM + TM | 2.60±0.00 | 3.18±0.00 | 0.00±0.00 | 1.66±0.00 | 3.27±0.64 | 6.62±0.00 | 7.28±0.00 |
| MM | 2.60±0.00 | 3.18±0.00 | 0.00±0.00 | 1.66±0.00 | 3.77±0.08 | 6.62±0.00 | 7.28±0.00 |
| ParetoFlow | 4.09±0.06 | 4.78±0.07 | 4.80±0.02 | **5.04±0.06** | 4.96±0.15 | 10.56±0.14 | 11.03±0.38 |
| PCD (ours) | **4.35±0.00** | **4.89±0.01** | **4.84±0.00** | 4.88±0.02 | **5.06±0.03** | **11.35±0.06** | **19.94±5.75** |

Table 13: Average 100th percentile hypervolume for RE tasks (part 2)

| Method | RE33 | RE34 | RE35 | RE36 | RE37 | RE41 | RE42 | RE61 |
|---|---|---|---|---|---|---|---|---|
| $\mathcal{D}$(best) | 10.56 | 9.30 | 10.08 | 7.61 | 5.57 | **18.27** | 14.52 | 97.49 |
| E2E + GN | 0.00±0.00 | 6.31±0.00 | 0.00±0.00 | 0.00±0.00 | 3.34±0.00 | 15.13±0.00 | 0.00±0.00 | 45.11±0.00 |
| E2E + PC | 0.00±0.00 | 6.31±0.00 | 0.00±0.00 | 0.00±0.00 | 3.34±0.00 | 15.12±0.00 | 0.00±0.00 | 45.11±0.00 |
| E2E | 0.00±0.00 | 6.31±0.00 | 0.00±0.00 | 0.00±0.00 | 3.34±0.00 | 15.12±0.00 | 0.00±0.00 | 45.11±0.00 |
| MOBO | 0.00±0.00 | 4.02±0.04 | 0.00±0.00 | 0.00±0.00 | 2.80±0.00 | 13.03±0.03 | 0.00±0.00 | 0.00±0.00 |
| MH + GN | 0.00±0.00 | 5.89±0.38 | 0.00±0.00 | 0.00±0.00 | 3.33±0.00 | 14.99±0.12 | 0.00±0.00 | 45.11±0.00 |
| MH + PC | 0.00±0.00 | 6.31±0.00 | 0.00±0.00 | 0.00±0.00 | 3.34±0.00 | 15.12±0.00 | 0.00±0.00 | 45.11±0.00 |
| MH | 0.00±0.00 | 6.31±0.00 | 0.00±0.00 | 0.00±0.00 | 3.34±0.00 | 15.12±0.00 | 0.00±0.00 | 45.11±0.00 |
| MM + COMs | 0.00±0.00 | 6.31±0.01 | 0.00±0.00 | 0.00±0.00 | 3.33±0.00 | 15.10±0.04 | 0.00±0.00 | 45.11±0.00 |
| MM + ICT | 0.00±0.00 | 6.31±0.00 | 0.00±0.00 | 0.00±0.00 | 3.34±0.00 | 15.13±0.00 | 0.00±0.00 | 45.11±0.00 |
| MM + IOM | 0.00±0.00 | 6.31±0.00 | 0.00±0.00 | 0.00±0.00 | 3.34±0.00 | 15.13±0.00 | 0.00±0.00 | 45.11±0.00 |
| MM + TM | 0.00±0.00 | 6.31±0.01 | 0.00±0.00 | 0.00±0.00 | 3.34±0.00 | 15.12±0.00 | 0.00±0.00 | 45.11±0.00 |
| MM | 0.00±0.00 | 6.31±0.00 | 0.00±0.00 | 0.00±0.00 | 3.34±0.00 | 15.12±0.00 | 0.00±0.00 | 45.11±0.00 |
| ParetoFlow | 10.84±0.24 | **11.69±1.33** | **11.12±0.43** | **8.67±0.17** | **6.51±0.58** | 18.20±0.41 | **54.44±22.78** | **112.08±11.62** |
| PCD (ours) | **10.97±0.09** | 10.17±0.04 | 10.87±0.04 | 8.16±0.11 | 5.97±0.01 | 18.14±0.02 | 15.49±0.03 | N/A |

Table 14: Average 100th percentile hypervolume for MONAS tasks (part 1)

| Method | C10/MOP1 | C10/MOP2 | C10/MOP3 | C10/MOP4 | C10/MOP5 | C10/MOP6 | C10/MOP7 | C10/MOP8 | C10/MOP9 |
|---|---|---|---|---|---|---|---|---|---|
| $\mathcal{D}$(best) | 4.47 | 10.47 | 9.21 | 18.62 | 45.11 | 103.78 | 422.29 | 4.70 | 10.79 |
| E2E + GN | 4.48±0.02 | 10.48±0.07 | 10.05±0.02 | 20.84±0.20 | 49.03±0.14 | 102.42±4.75 | 445.77±20.86 | 4.61±0.08 | 10.72±0.16 |
| E2E + PC | 4.47±0.01 | 10.52±0.07 | 10.18±0.01 | 21.64±0.13 | **49.22±0.09** | 105.99±0.49 | 492.21±1.55 | 4.65±0.08 | 10.46±0.29 |
| E2E | 4.47±0.00 | 10.55±0.05 | 10.19±0.01 | 21.70±0.13 | 48.87±0.05 | 106.17±0.36 | **494.92±1.70** | 4.77±0.03 | 10.64±0.15 |
| MOBO | 4.46±0.01 | 10.51±0.06 | 9.57±0.00 | 20.80±0.01 | 49.01±0.02 | **106.54±0.04** | 494.82±0.77 | 4.85±0.02 | 11.05±0.05 |
| MH + GN | 4.47±0.01 | 10.45±0.01 | 9.53±0.26 | 19.50±1.33 | 48.47±0.41 | 82.40±4.76 | 427.04±9.41 | 4.41±0.16 | 10.02±0.09 |
| MH + PC | 4.46±0.00 | 10.53±0.04 | 10.19±0.02 | 21.67±0.08 | 49.13±0.12 | 106.03±0.59 | 492.27±2.51 | 4.54±0.06 | 10.68±0.22 |
| MH | 4.47±0.01 | 10.47±0.03 | 10.17±0.02 | 21.26±0.45 | 49.18±0.07 | 105.88±0.64 | 490.43±2.20 | 4.72±0.07 | 11.10±0.27 |
| MM + COMs | 4.49±0.01 | 10.49±0.03 | 9.95±0.06 | 20.84±0.33 | 47.86±0.75 | 105.12±0.56 | 484.34±3.79 | **4.90±0.07** | 11.04±0.13 |
| MM + ICT | 4.46±0.02 | 10.52±0.05 | 10.00±0.07 | 20.85±0.19 | 48.98±0.15 | 106.08±0.35 | 494.18±1.96 | 4.62±0.10 | 10.34±0.18 |
| MM + IOM | 4.46±0.01 | 10.42±0.03 | 10.12±0.04 | 21.71±0.04 | 49.20±0.07 | 106.28±0.29 | 494.33±0.69 | 4.89±0.06 | **11.50±0.05** |
| MM + TM | 4.48±0.01 | 10.46±0.09 | 10.14±0.03 | 21.36±0.26 | 49.06±0.14 | 106.42±0.17 | 491.82±1.86 | 4.57±0.07 | 10.22±0.34 |
| MM | 4.46±0.01 | 10.49±0.06 | **10.20±0.01** | **21.77±0.09** | 48.96±0.07 | 105.33±0.43 | 494.49±0.80 | 4.82±0.05 | 10.99±0.26 |
| ParetoFlow | 4.48±0.02 | 10.45±0.01 | 9.48±0.15 | 19.50±0.39 | 44.25±2.48 | 104.19±0.70 | 435.03±20.16 | 4.65±0.08 | 10.79±0.08 |
| PCD (ours) | **4.50±0.01** | **10.59±0.04** | 9.75±0.06 | 20.43±0.12 | 41.88±0.76 | 105.82±0.87 | N/A | 4.79±0.02 | 10.99±0.19 |

Table 15: Average 100th percentile hypervolume for MONAS tasks (part 2)

| Method | IN-1K/MOP1 | IN-1K/MOP2 | IN-1K/MOP3 | IN-1K/MOP4 | IN-1K/MOP5 | IN-1K/MOP6 | IN-1K/MOP7 | IN-1K/MOP8 | IN-1K/MOP9 |
|---|---|---|---|---|---|---|---|---|---|
| $\mathcal{D}$(best) | 4.47 | **4.53** | 10.06 | 4.15 | 4.30 | 9.15 | 3.94 | 9.28 | 18.07 |
| E2E + GN | 4.63±0.04 | 4.16±0.08 | 9.93±0.07 | 4.44±0.05 | 4.46±0.07 | 9.71±0.22 | 4.40±0.09 | 9.17±0.10 | 18.84±0.30 |
| E2E + PC | **4.70±0.04** | 4.07±0.10 | 10.21±0.03 | 4.50±0.10 | 4.53±0.09 | 9.88±0.14 | 4.34±0.12 | 9.55±0.13 | 19.52±0.22 |
| E2E | 4.70±0.05 | 4.05±0.09 | 10.25±0.03 | 4.47±0.08 | 4.62±0.02 | 10.03±0.18 | 4.52±0.10 | 9.72±0.08 | 20.12±0.13 |
| MOBO | 4.34±0.03 | 4.27±0.04 | 9.97±0.04 | 4.43±0.10 | 4.52±0.05 | 9.84±0.11 | 3.94±0.03 | 9.29±0.10 | 19.73±0.09 |
| MH + GN | 4.47±0.16 | 4.06±0.10 | 7.43±1.19 | 4.26±0.18 | 4.48±0.11 | 9.47±0.43 | 4.31±0.11 | 7.17±1.68 | 14.48±1.25 |
| MH + PC | 4.64±0.02 | 4.16±0.07 | 10.07±0.10 | 4.40±0.05 | 4.59±0.07 | 9.78±0.25 | 4.45±0.02 | 9.40±0.13 | 19.28±0.32 |
| MH | 4.65±0.03 | 3.81±0.22 | 10.17±0.05 | 4.58±0.04 | 4.61±0.02 | 10.07±0.10 | **4.59±0.04** | 9.60±0.06 | 19.69±0.18 |
| MM + COMs | 4.64±0.07 | 4.34±0.02 | 10.19±0.01 | 4.29±0.20 | 4.51±0.11 | 9.02±0.86 | 4.15±0.08 | 9.73±0.03 | 20.17±0.09 |
| MM + ICT | 4.69±0.02 | 4.09±0.18 | 10.06±0.06 | 4.42±0.07 | 4.57±0.10 | 9.76±0.18 | 4.40±0.09 | 8.93±0.19 | 18.71±0.30 |
| MM + IOM | 4.67±0.02 | 4.09±0.17 | 10.22±0.02 | 4.20±0.45 | 4.55±0.12 | 9.56±0.06 | 4.40±0.13 | **9.82±0.03** | **20.39±0.06** |
| MM + TM | 4.61±0.03 | 4.11±0.12 | 10.07±0.03 | 4.48±0.03 | 4.57±0.03 | 9.73±0.15 | 4.40±0.32 | 9.39±0.16 | 19.42±0.17 |
| MM | 4.69±0.05 | 4.18±0.05 | **10.28±0.06** | **4.59±0.09** | **4.64±0.05** | **10.14±0.17** | 4.56±0.07 | 9.65±0.08 | 19.99±0.32 |
| ParetoFlow | 4.46±0.01 | 4.35±0.06 | 9.96±0.08 | 4.28±0.07 | 4.45±0.08 | 9.43±0.19 | 3.81±0.08 | 9.32±0.09 | 18.95±0.52 |
| PCD (ours) | 4.63±0.05 | 4.43±0.02 | 10.20±0.03 | 4.36±0.07 | 4.56±0.02 | 9.76±0.11 | 4.22±0.05 | 9.38±0.03 | 18.83±0.31 |

## A.13 50TH PERCENTILE RESULTS

Tables 25, 26, 27, 28, 29, 30, 31 and 32 showcase the per task 50th percentile hypervolume (↑) across all considered tasks. In each task, the best performing model is **bolded**, while the runner up is underlined.

Table 16: Average 100th percentile hypervolume for MORL tasks

| Method | MO-Hopper-v2 | MO-Swimmer-v2 |
|---|---|---|
| $\mathcal{D}$(best) | **6.33** | **3.70** |
| E2E + GN | 5.94±0.33 | 3.65±0.02 |
| E2E + PC | 6.22±0.15 | 3.66±0.00 |
| E2E | 5.98±0.34 | 1.68±0.75 |
| MOBO | 4.75±0.00 | 0.86±0.00 |
| MH + GN | 5.73±0.26 | 3.49±0.16 |
| MH + PC | 4.99±0.34 | 3.42±0.36 |
| MH | 5.64±0.54 | 3.20±0.98 |
| MM + COMs | 6.14±0.15 | 3.68±0.02 |
| MM + ICT | 5.67±0.26 | 3.12±1.11 |
| MM + IOM | 4.95±0.44 | 1.95±0.53 |
| MM + TM | 5.93±0.30 | 3.55±0.05 |
| MM | 5.99±0.19 | 1.61±0.58 |
| ParetoFlow | 5.92±0.05 | 3.50±0.10 |
| PCD (ours) | 5.95±0.18 | 3.69±0.11 |

Table 17: Average 75th percentile hypervolume for synthetic tasks (part 1)

| Method | ZDT1 | ZDT2 | ZDT3 | ZDT4 | ZDT6 |
|---|---|---|---|---|---|
| $\mathcal{D}$(best) | 4.17 | 4.68 | 5.15 | **5.46** | **4.61** |
| E2E + GN | 2.60±0.24 | 2.77±0.00 | 3.17±0.00 | 4.59±0.15 | 3.79±0.05 |
| E2E + PC | 2.81±0.00 | 3.17±0.00 | 3.16±0.20 | 4.54±0.17 | 3.78±0.09 |
| E2E | 2.81±0.00 | 3.17±0.00 | 3.48±0.13 | 4.43±0.23 | 3.82±0.04 |
| MOBO | 2.36±0.01 | 2.74±0.01 | 3.01±0.05 | 4.60±0.06 | 3.74±0.00 |
| MH + GN | 2.78±0.02 | 3.17±0.00 | 3.54±0.00 | 4.41±0.15 | 3.73±0.03 |
| MH + PC | 2.26±0.09 | 2.68±0.01 | 3.27±0.22 | 4.55±0.15 | 3.76±0.04 |
| MH | 2.81±0.00 | 3.17±0.00 | 3.52±0.03 | 4.55±0.09 | 3.76±0.00 |
| MM + COMs | 2.79±0.02 | 3.17±0.00 | 3.54±0.00 | 4.49±0.10 | 3.80±0.05 |
| MM + ICT | 2.80±0.00 | 3.17±0.00 | 3.54±0.00 | 4.48±0.13 | 3.80±0.06 |
| MM + IOM | 2.80±0.00 | 3.17±0.00 | 3.06±0.09 | 4.53±0.06 | 3.82±0.04 |
| MM + TM | 2.78±0.05 | 3.17±0.00 | 3.54±0.00 | 4.47±0.20 | 3.82±0.04 |
| MM | 2.81±0.00 | 3.17±0.00 | 3.53±0.02 | 4.57±0.18 | 3.76±0.00 |
| ParetoFlow | 4.11±0.07 | 5.45±0.25 | **5.21±0.12** | 4.77±0.12 | 4.40±0.04 |
| PCD (ours) | **4.25±0.10** | **5.84±0.10** | 4.09±0.16 | 4.91±0.03 | 4.31±0.08 |

Table 18: Average 75th percentile hypervolume for synthetic tasks (part 2)

| Method | DTLZ1 | DTLZ7 | OMNITEST | VLMOP1 | VLMOP2 | VLMOP3 |
|---|---|---|---|---|---|---|
| $\mathcal{D}$(best) | 10.60±0.00 | 8.56±0.00 | 4.53±0.00 | **0.41±0.00** | 1.78±0.00 | 45.65±0.00 |
| E2E + GN | 10.63±0.00 | 8.86±0.00 | 3.78±0.16 | 0.16±0.01 | 2.17±0.21 | 39.87±1.22 |
| E2E + PC | 10.61±0.03 | 8.86±0.00 | 4.67±0.07 | 0.13±0.04 | 3.34±0.01 | 42.44±0.01 |
| E2E | 10.28±0.22 | 8.86±0.00 | 4.67±0.08 | 0.14±0.03 | 3.35±0.01 | 42.45±0.00 |
| MOBO | 10.07±0.09 | 3.89±0.14 | 4.61±0.01 | 0.16±0.01 | 1.73±0.03 | 38.98±0.14 |
| MH + GN | 10.61±0.03 | 8.85±0.01 | 3.44±0.32 | 0.06±0.03 | 2.51±0.09 | 29.57±5.85 |
| MH + PC | 10.63±0.00 | 8.86±0.00 | 4.31±0.24 | 0.16±0.06 | 2.82±0.21 | 38.20±2.94 |
| MH | 10.62±0.00 | 8.86±0.00 | 4.30±0.50 | 0.20±0.05 | 3.31±0.01 | 42.44±0.00 |
| MM + COMs | 10.30±0.04 | 8.86±0.00 | 4.25±0.32 | 0.18±0.12 | 1.65±0.02 | 41.72±1.49 |
| MM + ICT | 10.55±0.07 | 8.86±0.00 | 4.44±0.17 | 0.11±0.02 | 3.34±0.01 | 42.39±0.02 |
| MM + IOM | 10.63±0.00 | 8.86±0.00 | 4.72±0.03 | 0.29±0.00 | 3.31±0.04 | 42.44±0.00 |
| MM + TM | 10.61±0.02 | 8.85±0.01 | 4.18±0.53 | 0.14±0.02 | 3.35±0.01 | 42.34±0.06 |
| MM | 10.61±0.02 | 8.86±0.00 | 4.70±0.06 | 0.09±0.01 | 3.34±0.01 | 42.45±0.01 |
| ParetoFlow | 10.83±0.36 | 10.32±0.43 | **4.78±0.00** | N/A | **4.21±0.03** | **45.90±0.02** |
| PCD (ours) | **13.59±0.78** | **11.26±0.15** | 4.34±0.05 | 0.01±0.01 | 2.12±0.04 | 45.81±0.04 |

Table 19: Average 75th percentile hypervolume for scientific design tasks

| Method | Molecule | RFP | Regex | Zinc |
|---|---|---|---|---|
| $\mathcal{D}$(best) | **2.91±0.00** | 7.11±0.00 | 3.96±0.00 | 4.52±0.00 |
| E2E + GN | 1.55±0.08 | 7.14±0.03 | 3.16±0.38 | 4.52±0.05 |
| E2E + PC | 2.47±0.07 | 7.15±0.03 | 4.15±0.77 | 4.54±0.05 |
| E2E | 2.41±0.01 | 7.17±0.02 | 2.99±0.00 | 4.57±0.04 |
| MOBO | 1.18±0.66 | **7.21±0.01** | **5.35±0.52** | 4.31±0.05 |
| MH + GN | 2.36±0.07 | 7.15±0.03 | 3.33±0.46 | 4.52±0.07 |
| MH + PC | 1.73±1.04 | 7.11±0.04 | 4.21±0.41 | 4.54±0.05 |
| MH | 2.41±0.01 | 7.15±0.04 | 2.99±0.00 | 4.56±0.05 |
| MM + COMs | 2.53±0.03 | 7.14±0.04 | 4.52±0.45 | 4.59±0.05 |
| MM + ICT | 2.50±0.07 | 7.18±0.01 | 3.53±0.81 | 4.60±0.05 |
| MM + IOM | 0.00±0.00 | 7.18±0.01 | 4.46±0.82 | 4.53±0.03 |
| MM + TM | 1.83±0.45 | 7.13±0.05 | 3.73±1.00 | 4.53±0.06 |
| MM | 2.50±0.07 | 7.16±0.03 | 2.99±0.00 | **4.60±0.04** |
| ParetoFlow | 1.75±0.25 | 7.04±0.11 | 5.03±0.47 | 4.40±0.03 |
| PCD (ours) | 1.84±0.09 | 7.19±0.02 | 4.14±0.25 | 4.50±0.02 |

Table 20: Average 75th percentile hypervolume for RE tasks (part 1)

| Method | RE21 | RE22 | RE23 | RE24 | RE25 | RE31 | RE32 |
|---|---|---|---|---|---|---|---|
| $\mathcal{D}$(best) | 4.10 | **4.78** | 4.75 | 4.60 | 4.79 | 10.60 | 10.56 |
| E2E + GN | 2.57±0.00 | 0.00±0.00 | 0.00±0.00 | 0.00±0.00 | 0.00±0.00 | 0.00±0.00 | 0.00±0.00 |
| E2E + PC | 2.57±0.00 | 0.00±0.00 | 0.00±0.00 | 0.00±0.00 | 0.00±0.00 | 1.00±2.24 | 2.30±2.20 |
| E2E | 2.57±0.00 | 0.00±0.00 | 0.00±0.00 | 0.00±0.00 | 0.00±0.00 | 2.49±2.58 | 3.15±2.05 |
| MOBO | 1.94±0.01 | 0.00±0.00 | 0.00±0.00 | 0.00±0.00 | 0.00±0.00 | 0.00±0.00 | 0.00±0.00 |
| MH + GN | 2.40±0.32 | 0.00±0.00 | 0.00±0.00 | 0.00±0.00 | 0.00±0.00 | 2.30±3.19 | 0.99±1.35 |
| MH + PC | 2.57±0.00 | 0.00±0.00 | 0.00±0.00 | 0.00±0.00 | 0.00±0.00 | 2.41±3.32 | 0.61±1.36 |
| MH | 2.57±0.00 | 0.00±0.00 | 0.00±0.00 | 0.00±0.00 | 0.00±0.00 | 5.32±0.94 | 3.12±2.10 |
| MM + COMs | 2.56±0.01 | 0.00±0.00 | 0.00±0.00 | 0.00±0.00 | 0.00±0.00 | 0.00±0.00 | 1.41±1.25 |
| MM + ICT | 2.57±0.00 | 0.00±0.00 | 0.00±0.00 | 0.00±0.00 | 0.00±0.00 | 5.00±1.43 | 1.32±1.81 |
| MM + IOM | 2.57±0.00 | 0.00±0.00 | 0.00±0.00 | 0.00±0.00 | 0.00±0.00 | 3.17±2.06 | 1.13±1.57 |
| MM + TM | 2.57±0.00 | 0.00±0.00 | 0.00±0.00 | 0.00±0.00 | 0.00±0.00 | 5.72±0.12 | 1.13±1.57 |
| MM | 2.57±0.00 | 0.00±0.00 | 0.00±0.00 | 0.00±0.00 | 0.00±0.00 | 2.32±3.18 | 2.76±0.91 |
| ParetoFlow | 4.05±0.05 | 4.72±0.08 | 4.75±0.00 | 4.51±0.04 | 4.81±0.02 | 10.31±0.08 | **11.00±0.38** |
| PCD (ours) | **4.28±0.01** | 4.74±0.03 | **4.83±0.00** | **4.86±0.01** | **4.86±0.01** | **10.74±0.13** | 10.67±0.03 |

Table 21: Average 75th percentile hypervolume for RE tasks (part 2)

| Method | RE33 | RE34 | RE35 | RE36 | RE37 | RE41 | RE42 | RE61 |
|---|---|---|---|---|---|---|---|---|
| $\mathcal{D}$(best) | 10.56 | 9.30 | 10.08 | 7.61 | 5.57 | **18.27** | 14.52 | 97.49 |
| E2E + GN | 0.00±0.00 | 6.29±0.01 | 0.00±0.00 | 0.00±0.00 | 3.33±0.00 | 14.78±0.16 | 0.00±0.00 | 45.09±0.01 |
| E2E + PC | 0.00±0.00 | 6.28±0.02 | 0.00±0.00 | 0.00±0.00 | 3.33±0.00 | 14.84±0.11 | 0.00±0.00 | 45.09±0.02 |
| E2E | 0.00±0.00 | 6.27±0.02 | 0.00±0.00 | 0.00±0.00 | 3.33±0.00 | 14.81±0.10 | 0.00±0.00 | 45.08±0.02 |
| MOBO | 0.00±0.00 | 3.80±0.10 | 0.00±0.00 | 0.00±0.00 | 2.79±0.00 | 12.99±0.06 | 0.00±0.00 | N/A |
| MH + GN | 0.00±0.00 | 5.36±0.66 | 0.00±0.00 | 0.00±0.00 | 3.28±0.04 | 14.23±0.71 | 0.00±0.00 | 45.08±0.01 |
| MH + PC | 0.00±0.00 | 6.27±0.02 | 0.00±0.00 | 0.00±0.00 | 3.33±0.00 | 14.78±0.14 | 0.00±0.00 | 45.08±0.01 |
| MH | 0.00±0.00 | 6.28±0.01 | 0.00±0.00 | 0.00±0.00 | 3.33±0.00 | 14.81±0.11 | 0.00±0.00 | 45.09±0.01 |
| MM + COMs | 0.00±0.00 | 6.29±0.01 | 0.00±0.00 | 0.00±0.00 | 3.33±0.01 | 14.84±0.18 | 0.00±0.00 | 45.11±0.00 |
| MM + ICT | 0.00±0.00 | 6.27±0.02 | 0.00±0.00 | 0.00±0.00 | 3.33±0.00 | 14.70±0.12 | 0.00±0.00 | 45.09±0.02 |
| MM + IOM | 0.00±0.00 | 6.29±0.02 | 0.00±0.00 | 0.00±0.00 | 3.33±0.00 | 14.85±0.12 | 0.00±0.00 | 45.09±0.02 |
| MM + TM | 0.00±0.00 | 6.28±0.02 | 0.00±0.00 | 0.00±0.00 | 3.33±0.00 | 14.78±0.09 | 0.00±0.00 | 45.10±0.00 |
| MM | 0.00±0.00 | 6.29±0.01 | 0.00±0.00 | 0.00±0.00 | 3.33±0.00 | 14.75±0.08 | 0.00±0.00 | 45.08±0.01 |
| ParetoFlow | 10.84±0.24 | **10.75±1.13** | **11.12±0.43** | **8.01±0.11** | **6.19±0.71** | 17.88±0.49 | **35.94±7.81** | **111.35±11.74** |
| PCD (ours) | **10.86±0.07** | 9.96±0.04 | 10.82±0.01 | 7.92±0.04 | 5.96±0.02 | 18.05±0.07 | 15.42±0.04 | N/A |

Table 22: Average 75th percentile hypervolume for MONAS tasks (part 1)

| Method | C10/MOP1 | C10/MOP2 | C10/MOP3 | C10/MOP4 | C10/MOP5 | C10/MOP6 | C10/MOP7 | C10/MOP8 | C10/MOP9 |
|---|---|---|---|---|---|---|---|---|---|
| $\mathcal{D}$(best) | **4.47** | **10.47** | 9.21 | 18.62 | 45.11 | 103.78 | 422.29 | 4.70 | 10.79 |
| E2E + GN | 4.47±0.02 | 10.45±0.03 | 9.92±0.15 | 20.65±0.19 | 48.35±0.48 | 92.88±0.57 | 430.13±1.84 | 4.37±0.10 | 10.14±0.14 |
| E2E + PC | 4.44±0.01 | 10.47±0.02 | 10.11±0.01 | 21.29±0.21 | **49.01±0.11** | 104.93±0.53 | 491.16±1.25 | 4.50±0.11 | 10.19±0.27 |
| E2E | 4.45±0.01 | 10.46±0.02 | 10.12±0.01 | 21.33±0.25 | 48.76±0.04 | 104.96±2.03 | **493.24±2.36** | 4.56±0.10 | 10.24±0.18 |
| MOBO | 4.44±0.01 | 10.43±0.02 | 9.41±0.06 | 20.55±0.15 | 48.62±0.09 | **105.83±0.05** | 491.45±2.98 | 4.74±0.04 | 10.54±0.27 |
| MH + GN | 4.45±0.01 | 10.44±0.01 | 9.37±0.43 | 19.30±1.25 | 42.59±3.88 | 75.36±4.92 | 394.79±21.78 | 4.18±0.17 | 9.59±0.24 |
| MH + PC | 4.44±0.01 | 10.46±0.02 | 10.11±0.04 | 21.20±0.17 | 48.73±0.19 | 103.54±2.16 | 485.73±2.48 | 4.36±0.06 | 10.38±0.22 |
| MH | 4.45±0.00 | 10.45±0.01 | 10.11±0.02 | 21.02±0.40 | 49.01±0.03 | 105.10±0.57 | 485.29±2.33 | 4.49±0.06 | 10.68±0.27 |
| MM + COMs | 4.45±0.02 | 10.45±0.02 | 9.78±0.07 | 20.53±0.42 | 45.40±2.11 | 103.29±0.91 | 479.91±3.12 | 4.77±0.06 | 10.81±0.07 |
| MM + ICT | 4.44±0.01 | 10.45±0.01 | 9.87±0.08 | 20.59±0.30 | 47.25±1.20 | 105.10±0.41 | 488.60±3.26 | 4.45±0.10 | 9.78±0.18 |
| MM + IOM | 4.45±0.01 | 10.41±0.02 | 10.05±0.06 | **21.46±0.21** | 48.85±0.36 | 105.74±0.57 | 493.09±1.78 | **4.79±0.12** | **11.33±0.12** |
| MM + TM | 4.45±0.02 | 10.42±0.05 | 10.04±0.11 | 21.02±0.48 | 48.59±0.49 | 105.29±0.20 | 484.79±7.15 | 4.46±0.09 | 9.86±0.31 |
| MM | 4.44±0.02 | 10.45±0.00 | **10.14±0.03** | 21.24±0.28 | 48.75±0.02 | 104.46±0.87 | 492.55±1.95 | 4.66±0.09 | 10.65±0.40 |
| ParetoFlow | 4.44±0.04 | 10.44±0.01 | 9.35±0.17 | 19.44±0.37 | 41.44±2.00 | 102.22±1.30 | 429.75±24.23 | 4.48±0.12 | 10.58±0.15 |
| PCD (ours) | 4.43±0.01 | 10.43±0.03 | 9.72±0.09 | 20.25±0.19 | 40.63±0.44 | 104.49±1.10 | N/A | 4.67±0.02 | 10.66±0.13 |

Table 23: Average 75th percentile hypervolume for MONAS tasks (part 2)

| Method | IN-1K/MOP1 | IN-1K/MOP2 | IN-1K/MOP3 | IN-1K/MOP4 | IN-1K/MOP5 | IN-1K/MOP6 | IN-1K/MOP7 | IN-1K/MOP8 | IN-1K/MOP9 |
|---|---|---|---|---|---|---|---|---|---|
| $\mathcal{D}$(best) | 4.47 | **4.53** | 10.06 | 4.15 | 4.30 | 9.15 | 3.94 | 9.28 | 18.07 |
| E2E + GN | 4.52±0.09 | 4.07±0.10 | 9.87±0.09 | 4.33±0.08 | 4.29±0.10 | 9.29±0.33 | 4.21±0.11 | 9.09±0.16 | 18.78±0.30 |
| E2E + PC | 4.68±0.04 | 3.92±0.17 | 10.19±0.02 | 4.37±0.09 | 4.42±0.07 | 9.63±0.16 | 4.20±0.15 | 9.45±0.12 | 19.33±0.20 |
| E2E | **4.69±0.06** | 3.79±0.11 | 10.24±0.02 | 4.40±0.06 | 4.53±0.03 | 9.77±0.19 | 4.47±0.11 | 9.67±0.08 | 19.94±0.14 |
| MOBO | 4.31±0.03 | 4.21±0.03 | 9.96±0.03 | 4.35±0.06 | 4.44±0.05 | 9.53±0.13 | 3.79±0.10 | 9.25±0.07 | 19.65±0.11 |
| MH + GN | 4.39±0.16 | 3.87±0.22 | 7.06±1.63 | 4.18±0.21 | 4.33±0.23 | 9.20±0.43 | 4.16±0.12 | 7.05±1.73 | 14.45±1.23 |
| MH + PC | 4.62±0.02 | 4.09±0.09 | 10.04±0.12 | 4.36±0.06 | 4.46±0.05 | 9.54±0.24 | 4.36±0.08 | 9.22±0.14 | 19.17±0.40 |
| MH | 4.63±0.02 | 3.66±0.26 | 10.15±0.06 | 4.49±0.02 | 4.52±0.04 | 9.90±0.08 | **4.54±0.06** | 9.53±0.07 | 19.60±0.23 |
| MM + COMs | 4.61±0.07 | 4.16±0.09 | 10.17±0.01 | 4.16±0.23 | 4.42±0.12 | 8.41±1.01 | 4.03±0.10 | 9.70±0.04 | 20.08±0.11 |
| MM + ICT | 4.67±0.03 | 3.91±0.21 | 10.02±0.08 | 4.31±0.05 | 4.47±0.09 | 9.51±0.14 | 4.21±0.07 | 8.85±0.16 | 18.48±0.23 |
| MM + IOM | 4.64±0.03 | 3.91±0.25 | 10.20±0.03 | 4.06±0.46 | 4.43±0.08 | 9.33±0.03 | 4.25±0.09 | **9.79±0.03** | **20.25±0.12** |
| MM + TM | 4.57±0.02 | 3.94±0.25 | 10.02±0.05 | 4.36±0.03 | 4.50±0.04 | 9.45±0.13 | 4.29±0.34 | 9.37±0.16 | 19.32±0.18 |
| MM | 4.67±0.04 | 3.99±0.21 | **10.27±0.06** | **4.51±0.06** | **4.56±0.04** | **9.91±0.23** | 4.50±0.08 | 9.60±0.09 | 19.84±0.28 |
| ParetoFlow | 4.37±0.07 | 4.34±0.06 | 9.88±0.10 | 4.26±0.08 | 4.41±0.05 | 9.29±0.27 | 3.63±0.12 | 9.26±0.09 | 18.86±0.52 |
| PCD (ours) | 4.49±0.01 | 4.24±0.05 | 10.18±0.02 | 4.20±0.03 | 4.40±0.01 | 9.16±0.07 | 3.71±0.14 | 9.33±0.06 | 18.71±0.30 |

Table 24: Average 75th percentile hypervolume for MORL tasks

| Method | MO-Hopper-v2 | MO-Swimmer-v2 |
|---|---|---|
| $\mathcal{D}$(best) | **6.33** | **3.70** |
| E2E + GN | 5.57±0.04 | 3.60±0.03 |
| E2E + PC | 5.62±0.01 | 3.65±0.01 |
| E2E | 5.27±0.48 | 0.89±0.02 |
| MOBO | 4.75±0.00 | 0.86±0.00 |
| MH + GN | 5.20±0.42 | 1.08±0.34 |
| MH + PC | 4.75±0.00 | 1.04±0.30 |
| MH | 5.07±0.44 | 2.46±1.42 |
| MM + COMs | 5.63±0.02 | 3.64±0.04 |
| MM + ICT | 4.75±0.00 | 2.12±1.32 |
| MM + IOM | 4.75±0.00 | 1.18±0.36 |
| MM + TM | 4.75±0.00 | 1.11±0.20 |
| MM | 5.58±0.03 | 1.00±0.18 |
| ParetoFlow | 5.59±0.01 | 2.93±0.29 |
| PCD (ours) | 5.07±0.10 | 2.82±0.05 |

Table 25: Average 50th percentile hypervolume for synthetic tasks (part 1)

| Method | ZDT1 | ZDT2 | ZDT3 | ZDT4 | ZDT6 |
|---|---|---|---|---|---|
| $\mathcal{D}$(best) | **4.17** | 4.68 | **5.15** | **5.46** | **4.61** |
| E2E + GN | 2.34±0.00 | 2.76±0.01 | 3.16±0.00 | 4.49±0.13 | 3.74±0.02 |
| E2E + PC | 2.74±0.00 | 3.17±0.00 | 3.05±0.01 | 4.47±0.13 | 3.73±0.05 |
| E2E | 2.74±0.00 | 3.17±0.00 | 3.11±0.05 | 4.36±0.23 | 3.76±0.00 |
| MOBO | 2.35±0.02 | 2.71±0.03 | 2.92±0.04 | 4.43±0.03 | 3.74±0.00 |
| MH + GN | 2.64±0.17 | 3.17±0.00 | 3.19±0.01 | 4.37±0.15 | 3.73±0.03 |
| MH + PC | 2.25±0.09 | 2.68±0.01 | 3.14±0.19 | 4.49±0.13 | 3.74±0.01 |
| MH | 2.74±0.00 | 3.17±0.00 | 3.15±0.05 | 4.49±0.10 | 3.76±0.00 |
| MM + COMs | 2.74±0.00 | 3.17±0.00 | 3.07±0.00 | 4.42±0.12 | 3.70±0.05 |
| MM + ICT | 2.74±0.00 | 3.17±0.00 | 3.13±0.03 | 4.41±0.11 | 3.74±0.02 |
| MM + IOM | 2.74±0.00 | 3.17±0.00 | 2.97±0.01 | 4.48±0.07 | 3.76±0.00 |
| MM + TM | 2.60±0.18 | 3.17±0.00 | 3.20±0.01 | 4.44±0.19 | 3.71±0.05 |
| MM | 2.74±0.00 | 3.17±0.00 | 3.10±0.01 | 4.50±0.18 | 3.76±0.00 |
| ParetoFlow | 4.10±0.07 | 5.12±0.10 | 5.02±0.18 | 4.68±0.13 | 4.33±0.03 |
| PCD (ours) | 4.02±0.07 | **5.51±0.10** | 3.37±0.19 | 4.72±0.10 | 0.00±0.00 |

Table 26: Average 50th percentile hypervolume for synthetic tasks (part 2)

| Method | DTLZ1 | DTLZ7 | OMNITEST | VLMOP1 | VLMOP2 | VLMOP3 |
|---|---|---|---|---|---|---|
| $\mathcal{D}$(best) | 10.60 | 8.56 | 4.53 | **0.41** | 1.78 | 45.65 |
| E2E + GN | 10.31±0.05 | 8.77±0.00 | 3.37±0.14 | 0.00±0.00 | 1.99±0.20 | 37.46±0.92 |
| E2E + PC | 10.30±0.09 | 8.77±0.00 | 4.24±0.03 | 0.00±0.00 | 3.27±0.07 | 40.15±0.18 |
| E2E | 9.86±0.21 | 8.77±0.00 | 4.31±0.21 | 0.00±0.00 | 3.33±0.02 | 40.27±0.00 |
| MOBO | 9.81±0.09 | 3.81±0.17 | 4.25±0.04 | 0.00±0.00 | 1.64±0.02 | 38.98±0.14 |
| MH + GN | 10.10±0.13 | 8.78±0.00 | 3.26±0.21 | 0.04±0.02 | 2.02±0.05 | 29.19±5.40 |
| MH + PC | 10.31±0.05 | 8.77±0.00 | 4.01±0.21 | 0.09±0.07 | 2.54±0.21 | 35.96±1.79 |
| MH | 10.21±0.05 | 8.77±0.00 | 4.04±0.41 | 0.00±0.00 | 3.15±0.02 | 40.66±0.97 |
| MM + COMs | 10.06±0.08 | 8.78±0.01 | 4.01±0.33 | 0.06±0.13 | 1.55±0.03 | 39.03±1.41 |
| MM + ICT | 10.22±0.05 | 8.77±0.00 | 4.12±0.13 | 0.00±0.00 | 3.34±0.01 | 39.29±0.30 |
| MM + IOM | 10.33±0.05 | 8.77±0.00 | 4.29±0.03 | 0.19±0.05 | 3.25±0.05 | 40.23±0.06 |
| MM + TM | 10.24±0.04 | 8.77±0.01 | 3.93±0.46 | 0.00±0.00 | 3.34±0.01 | 38.88±0.25 |
| MM | 10.17±0.08 | 8.77±0.00 | 4.28±0.22 | 0.00±0.00 | 3.32±0.01 | 40.69±0.95 |
| ParetoFlow | 10.43±0.44 | 9.44±0.48 | **4.78±0.00** | N/A | **4.21±0.03** | 45.45±0.41 |
| PCD (ours) | **12.11±0.51** | **10.61±0.07** | 3.70±0.17 | 0.00±0.00 | 1.80±0.03 | **45.68±0.03** |

Table 27: Average 50th percentile hypervolume for scientific design tasks

| Method | Molecule | RFP | Regex | Zinc |
|---|---|---|---|---|
| $\mathcal{D}$(best) | **2.91±0.00** | 7.11±0.00 | 3.96±0.00 | **4.52±0.00** |
| E2E + GN | 1.24±0.70 | 6.96±0.13 | 2.99±0.00 | 4.42±0.02 |
| E2E + PC | 1.51±0.92 | 7.03±0.12 | 2.99±0.00 | 4.39±0.09 |
| E2E | 2.26±0.07 | 7.10±0.03 | 2.99±0.00 | 4.40±0.04 |
| MOBO | 0.00±0.00 | 7.13±0.00 | **5.05±0.67** | 4.03±0.13 |
| MH + GN | 1.23±0.69 | 6.97±0.05 | 3.16±0.38 | 4.34±0.09 |
| MH + PC | 0.45±1.01 | 7.05±0.01 | 3.76±0.47 | 4.33±0.12 |
| MH | 1.83±1.02 | 7.00±0.21 | 2.99±0.00 | 4.42±0.02 |
| MM + COMs | 2.31±0.00 | 7.07±0.04 | 3.44±0.64 | 4.43±0.02 |
| MM + ICT | 2.30±0.01 | 7.05±0.02 | 3.16±0.38 | 4.34±0.05 |
| MM + IOM | 0.00±0.00 | 7.08±0.05 | 2.99±0.00 | 4.39±0.03 |
| MM + TM | 1.19±0.66 | 7.08±0.04 | 3.36±0.82 | 4.35±0.07 |
| MM | 2.30±0.00 | 7.11±0.04 | 2.99±0.00 | 4.45±0.03 |
| ParetoFlow | 1.26±0.71 | 6.87±0.12 | 4.72±0.54 | 4.15±0.25 |
| PCD (ours) | 1.65±0.05 | **7.14±0.03** | 4.14±0.25 | 4.32±0.08 |

Table 28: Average 50th percentile hypervolume for RE tasks (part 1)

| Method | RE21 | RE22 | RE23 | RE24 | RE25 | RE31 | RE32 |
|---|---|---|---|---|---|---|---|
| $\mathcal{D}$(best) | 4.10 | **4.78** | 4.75 | 4.60 | 4.79 | **10.60** | 10.56 |
| E2E + GN | 2.54±0.00 | 0.00±0.00 | 0.00±0.00 | 0.00±0.00 | 0.00±0.00 | 0.00±0.00 | 0.00±0.00 |
| E2E + PC | 2.54±0.00 | 0.00±0.00 | 0.00±0.00 | 0.00±0.00 | 0.00±0.00 | 0.00±0.00 | 0.49±1.11 |
| E2E | 2.54±0.00 | 0.00±0.00 | 0.00±0.00 | 0.00±0.00 | 0.00±0.00 | 3.49±3.18 | 0.68±1.53 |
| MOBO | 1.90±0.02 | 0.00±0.00 | 0.00±0.00 | 0.00±0.00 | 0.00±0.00 | 0.00±0.00 | 0.00±0.00 |
| MH + GN | 2.33±0.29 | 0.00±0.00 | 0.00±0.00 | 0.00±0.00 | 0.00±0.00 | 1.16±2.60 | 0.00±0.00 |
| MH + PC | 2.54±0.00 | 0.00±0.00 | 0.00±0.00 | 0.00±0.00 | 0.00±0.00 | 0.00±0.00 | 0.00±0.00 |
| MH | 2.54±0.00 | 0.00±0.00 | 0.00±0.00 | 0.00±0.00 | 0.00±0.00 | 0.48±1.08 | 0.61±1.36 |
| MM + COMs | 2.32±0.02 | 0.00±0.00 | 0.00±0.00 | 0.00±0.00 | 0.00±0.00 | 0.00±0.00 | 0.20±0.46 |
| MM + ICT | 2.54±0.00 | 0.00±0.00 | 0.00±0.00 | 0.00±0.00 | 0.00±0.00 | 2.18±2.33 | 0.63±1.42 |
| MM + IOM | 2.54±0.00 | 0.00±0.00 | 0.00±0.00 | 0.00±0.00 | 0.00±0.00 | 2.17±2.16 | 0.00±0.00 |
| MM + TM | 2.54±0.00 | 0.00±0.00 | 0.00±0.00 | 0.00±0.00 | 0.00±0.00 | 2.32±3.18 | 0.00±0.00 |
| MM | 2.54±0.00 | 0.00±0.00 | 0.00±0.00 | 0.00±0.00 | 0.00±0.00 | 2.01±2.81 | 1.37±1.87 |
| ParetoFlow | 3.98±0.08 | 4.65±0.07 | 4.66±0.00 | 4.43±0.03 | 4.81±0.02 | 10.22±0.05 | **10.87±0.39** |
| PCD (ours) | **4.22±0.01** | 4.49±0.08 | **4.82±0.00** | **4.85±0.01** | **4.84±0.00** | 10.53±0.02 | 10.64±0.00 |

Table 29: Average 50th percentile hypervolume for RE tasks (part 2)

| Method | RE33 | RE34 | RE35 | RE36 | RE37 | RE41 | RE42 | RE61 |
|---|---|---|---|---|---|---|---|---|
| $\mathcal{D}$(best) | 10.56 | 9.30 | 10.08 | 7.61 | 5.57 | **18.27** | 14.52 | 97.49 |
| E2E + GN | 0.00±0.00 | 6.15±0.16 | 0.00±0.00 | 0.00±0.00 | 3.32±0.01 | 14.47±0.05 | 0.00±0.00 | 45.05±0.04 |
| E2E + PC | 0.00±0.00 | 6.22±0.01 | 0.00±0.00 | 0.00±0.00 | 3.31±0.00 | 14.44±0.16 | 0.00±0.00 | 45.06±0.03 |
| E2E | 0.00±0.00 | 6.21±0.00 | 0.00±0.00 | 0.00±0.00 | 3.31±0.00 | 14.49±0.12 | 0.00±0.00 | 45.07±0.03 |
| MOBO | 0.00±0.00 | 3.56±0.02 | 0.00±0.00 | 0.00±0.00 | 2.77±0.00 | 12.89±0.03 | 0.00±0.00 | N/A |
| MH + GN | 0.00±0.00 | 4.56±0.44 | 0.00±0.00 | 0.00±0.00 | 3.11±0.09 | 13.92±0.73 | 0.00±0.00 | 44.72±0.35 |
| MH + PC | 0.00±0.00 | 6.22±0.01 | 0.00±0.00 | 0.00±0.00 | 3.31±0.00 | 14.52±0.18 | 0.00±0.00 | 45.06±0.03 |
| MH | 0.00±0.00 | 6.21±0.00 | 0.00±0.00 | 0.00±0.00 | 3.31±0.00 | 14.55±0.18 | 0.00±0.00 | 45.09±0.01 |
| MM + COMs | 0.00±0.00 | 6.22±0.02 | 0.00±0.00 | 0.00±0.00 | 3.22±0.05 | 14.40±0.15 | 0.00±0.00 | 45.10±0.00 |
| MM + ICT | 0.00±0.00 | 6.20±0.06 | 0.00±0.00 | 0.00±0.00 | 3.31±0.00 | 14.47±0.10 | 0.00±0.00 | 45.06±0.03 |
| MM + IOM | 0.00±0.00 | 6.22±0.02 | 0.00±0.00 | 0.00±0.00 | 3.31±0.00 | 14.43±0.09 | 0.00±0.00 | 45.07±0.03 |
| MM + TM | 0.00±0.00 | 6.06±0.39 | 0.00±0.00 | 0.00±0.00 | 3.31±0.00 | 14.45±0.24 | 0.00±0.00 | 45.09±0.02 |
| MM | 0.00±0.00 | 6.21±0.00 | 0.00±0.00 | 0.00±0.00 | 3.31±0.00 | 14.57±0.04 | 0.00±0.00 | 45.06±0.02 |
| ParetoFlow | 10.58±0.19 | **9.64±0.79** | **10.85±0.37** | 7.41±0.24 | **5.89±0.51** | 17.77±0.50 | **20.08±4.86** | **106.15±6.82** |
| PCD (ours) | **10.80±0.08** | 9.47±0.09 | 10.57±0.03 | **7.71±0.03** | 5.83±0.02 | 17.98±0.03 | 15.19±0.07 | N/A |

Table 30: Average 50th percentile hypervolume for MONAS tasks (part 1)

| Method | C10/MOP1 | C10/MOP2 | C10/MOP3 | C10/MOP4 | C10/MOP5 | C10/MOP6 | C10/MOP7 | C10/MOP8 | C10/MOP9 |
|---|---|---|---|---|---|---|---|---|---|
| $\mathcal{D}$(best) | **4.47** | **10.47** | 9.21 | 18.62 | 45.11 | 103.78 | 422.29 | 4.70 | 10.79 |
| E2E + GN | 4.43±0.01 | 10.41±0.09 | 9.75±0.14 | 19.29±0.41 | 47.16±0.77 | 89.27±1.06 | 418.89±5.31 | 4.13±0.26 | 9.59±0.24 |
| E2E + PC | 4.41±0.02 | 10.43±0.02 | 10.01±0.04 | 20.06±0.24 | 48.40±0.42 | 103.72±0.24 | 486.25±3.00 | 4.37±0.07 | 9.87±0.33 |
| E2E | 4.41±0.03 | 10.40±0.06 | 10.01±0.07 | **20.47±0.25** | 48.43±0.24 | 103.81±2.47 | **489.60±3.03** | 4.43±0.08 | 9.96±0.14 |
| MOBO | 4.42±0.01 | 10.40±0.02 | 9.27±0.01 | 19.01±0.45 | 48.06±0.46 | **105.42±0.04** | 481.38±4.92 | 4.62±0.13 | 10.00±0.28 |
| MH + GN | 4.40±0.06 | 10.28±0.21 | 8.96±0.47 | 17.71±1.21 | 39.43±2.66 | 71.98±5.98 | 365.27±19.94 | 4.02±0.18 | 9.24±0.46 |
| MH + PC | 4.41±0.03 | 10.30±0.18 | 10.01±0.04 | 20.19±0.23 | 47.83±0.48 | 98.46±5.65 | 474.65±5.05 | 4.30±0.06 | 10.03±0.32 |
| MH | 4.42±0.01 | 10.43±0.01 | **10.03±0.03** | 19.98±0.20 | **48.89±0.02** | 102.76±2.49 | 476.25±4.32 | 4.39±0.04 | 10.36±0.37 |
| MM + COMs | 4.43±0.01 | 10.44±0.01 | 9.70±0.08 | 19.38±0.18 | 44.39±2.16 | 100.15±1.47 | 461.28±7.29 | 4.65±0.04 | 10.55±0.11 |
| MM + ICT | 4.42±0.01 | 10.43±0.01 | 9.69±0.08 | 18.63±1.00 | 45.65±1.26 | 104.04±0.49 | 477.05±3.01 | 4.35±0.09 | 9.47±0.11 |
| MM + IOM | 4.40±0.05 | 10.28±0.21 | 9.91±0.10 | 20.12±0.28 | 47.59±0.78 | 104.61±0.81 | 482.85±4.91 | **4.71±0.15** | **11.22±0.13** |
| MM + TM | 4.42±0.01 | 10.31±0.13 | 9.92±0.13 | 19.66±0.45 | 47.72±1.01 | 103.33±1.04 | 462.91±11.81 | 4.35±0.07 | 9.57±0.45 |
| MM | 4.38±0.02 | 10.44±0.01 | 10.02±0.05 | 20.11±0.33 | 48.58±0.12 | 102.62±0.95 | 485.56±3.94 | 4.53±0.14 | 10.33±0.48 |
| ParetoFlow | 4.37±0.06 | 10.40±0.04 | 9.00±0.42 | 18.26±0.78 | 39.00±0.60 | 100.52±2.07 | 407.48±21.71 | 4.30±0.08 | 10.37±0.10 |
| PCD (ours) | 4.41±0.01 | 10.26±0.13 | 9.31±0.11 | 18.40±0.38 | 39.61±0.40 | 101.57±1.19 | N/A | 4.60±0.01 | 10.34±0.13 |

Table 31: Average 50th percentile hypervolume for MONAS tasks (part 2)

| Method | IN-1K/MOP1 | IN-1K/MOP2 | IN-1K/MOP3 | IN-1K/MOP4 | IN-1K/MOP5 | IN-1K/MOP6 | IN-1K/MOP7 | IN-1K/MOP8 | IN-1K/MOP9 |
|---|---|---|---|---|---|---|---|---|---|
| $\mathcal{D}$(best) | 4.47 | **4.53** | 10.06 | 4.15 | 4.30 | 9.15 | 3.94 | 9.28 | 18.07 |
| E2E + GN | 4.41±0.16 | 3.77±0.30 | 9.44±0.53 | 4.10±0.38 | 3.42±0.14 | 8.62±0.96 | 3.71±0.34 | 8.59±0.26 | 17.68±0.59 |
| E2E + PC | **4.66±0.04** | 3.84±0.18 | 10.17±0.02 | 4.18±0.18 | 4.33±0.06 | 9.36±0.11 | 4.07±0.17 | 9.34±0.13 | 19.05±0.34 |
| E2E | 4.66±0.04 | 3.68±0.10 | 10.20±0.02 | 4.34±0.06 | 4.48±0.03 | 9.51±0.07 | 4.38±0.09 | 9.59±0.11 | 19.50±0.41 |
| MOBO | 4.30±0.02 | 4.16±0.01 | 9.89±0.09 | 4.30±0.08 | 4.38±0.05 | 9.27±0.13 | 3.65±0.13 | 9.14±0.14 | 19.36±0.33 |
| MH + GN | 4.29±0.25 | 3.80±0.20 | 6.72±1.49 | 4.05±0.26 | 4.25±0.23 | 8.95±0.46 | 4.02±0.11 | 6.72±1.68 | 13.26±1.41 |
| MH + PC | 4.61±0.01 | 3.97±0.16 | 9.94±0.18 | 4.24±0.05 | 4.39±0.08 | 9.28±0.20 | 4.24±0.08 | 9.15±0.16 | 18.80±0.56 |
| MH | 4.61±0.02 | 3.42±0.18 | 10.11±0.05 | 4.42±0.02 | 4.46±0.05 | 9.68±0.12 | **4.45±0.08** | 9.45±0.07 | 19.39±0.18 |
| MM + COMs | 4.58±0.09 | 4.06±0.14 | 10.13±0.02 | 3.95±0.27 | 4.35±0.15 | 7.46±0.89 | 3.91±0.08 | 9.60±0.08 | 19.90±0.11 |
| MM + ICT | 4.64±0.03 | 3.79±0.17 | 9.92±0.08 | 4.23±0.06 | 4.38±0.13 | 9.20±0.13 | 4.15±0.06 | 8.55±0.33 | 18.03±0.61 |
| MM + IOM | 4.59±0.06 | 3.76±0.35 | 10.17±0.03 | 3.93±0.45 | 4.31±0.05 | 9.06±0.11 | 4.18±0.06 | **9.74±0.06** | **20.10±0.21** |
| MM + TM | 4.52±0.03 | 3.82±0.23 | 9.97±0.07 | 4.29±0.02 | 4.42±0.06 | 9.23±0.10 | 4.22±0.35 | 9.16±0.27 | 18.87±0.33 |
| MM | 4.65±0.03 | 3.85±0.24 | **10.25±0.07** | **4.43±0.05** | **4.53±0.05** | **9.70±0.22** | 4.44±0.09 | 9.52±0.07 | 19.66±0.30 |
| ParetoFlow | 4.29±0.05 | 4.25±0.04 | 9.64±0.05 | 4.10±0.18 | 4.26±0.16 | 9.11±0.24 | 3.51±0.19 | 8.85±0.51 | 18.54±0.50 |
| PCD (ours) | 4.41±0.02 | 4.06±0.09 | 10.06±0.10 | 4.11±0.03 | 4.27±0.02 | 8.82±0.02 | 3.30±0.15 | 8.96±0.02 | 17.63±0.71 |

Table 32: Average 50th percentile hypervolume for MORL tasks

| Method | MO-Hopper-v2 | MO-Swimmer-v2 |
|---|---|---|
| $\mathcal{D}$(best) | **6.33** | **3.70** |
| E2E + GN | 5.06±0.44 | 1.02±0.17 |
| E2E + PC | 4.84±0.20 | 3.61±0.00 |
| E2E | 4.92±0.39 | 0.87±0.01 |
| MOBO | 4.75±0.00 | 0.86±0.00 |
| MH + GN | 4.90±0.35 | 0.97±0.19 |
| MH + PC | 4.75±0.00 | 0.95±0.13 |
| MH | 4.91±0.36 | 0.97±0.14 |
| MM + COMs | 5.32±0.34 | 3.50±0.26 |
| MM + ICT | 4.75±0.00 | 0.96±0.10 |
| MM + IOM | 4.75±0.00 | 1.00±0.23 |
| MM + TM | 4.75±0.00 | 0.93±0.08 |
| MM | 4.75±0.00 | 0.92±0.10 |
| ParetoFlow | 5.38±0.36 | 2.66±0.32 |
| PCD (ours) | 4.83±0.02 | 2.43±0.09 |

