# OpenReview forum: "Pareto-Conditioned Diffusion Models for Offline Multi-Objective Optimization"
_ICLR.cc/2026/Conference — ICLR 2026 Oral_

### Official Review · Reviewer_MLZS · 2025-10-25

**Soundness:** 3
**Presentation:** 3
**Contribution:** 3
**Rating:** 8
**Confidence:** 4

**Summary:**

The paper proposes Pareto-Conditioned Diffusion (PCD) for offline multi-objective optimization (MOO). Instead of learning explicit surrogates and optimizing them with MOEAs, PCD trains a conditional diffusion model over designs conditioned on objective vectors. Two key ingredients: (i) a multi-objective reweighting scheme that emphasizes bins with high-quality nondominated solutions; and (ii) a reference-direction conditioning set (NSGA-III-inspired with Riesz s-energy vectors + extrapolation + Gaussian jitter) to generate diverse target trade-offs.

**Strengths:**

The dominance-aware bin reweighting and NSGA-III-style conditioning looks intuitive and principled.

**Weaknesses:**

1. > Unlike prior methods (Xue et al., 2024; Yuan et al., 2025) that rely on complex, multi-stage pipelines of surrogate predictors
or separate optimization algorithms, PCD offers a more direct, end-to-end framework.

In my view, ParetoFlow is classifier guidance while this PCD is classifier-free guidance. I do not think the former is more complicated. Although paretoflow requires two stages: predictor training and generative model training, these are established pipelines for that and thus they should be simple instead of complex.

2. The propose conditioning search looks interesting. Have u compared the uniform vectors from Das-Dennis?

3. It looks weird in MORL that no methods can beat against the offline best? so what is the point of all methods?

4. Have u tried flow models instead of diffusion models?

5. I doubt whether we really need diffusion models for some tasks mentioned here. Diffusion models can be very useful it the data has meaningful patterns to learn like image, text, protein. In this context, some designs are pure math numbers. Do they have meaningful patterns?

**Questions:**

See Weakness.

---

> ### Author Response · Authors · 2025-11-20
> **Response to Reviewer MLZS (1/5)**
>
> We thank the reviewer for their insightful feedback and helpful suggestions to improve our paper. We address the specific questions below.
>
> ### **Q2:** The proposed conditioning search looks interesting. Have you compared the uniform vectors from Das-Dennis?
> - We have conducted this comparison in the newly added Appendix A.6.3 (Table 7). We observe that the simpler Das-Dennis method has a slight edge over Riesz s-Energy on most tasks, though the performance difference is marginal, demonstrating that PCD is robust to the choice of direction generation strategy. Additionally, we have added Figure 2 to the main text to better visualize the overall conditioning point generation procedure.
>
> | Method         | C10/MOP2       | MO-Hopper   | RE34          | Regex         | ZDT2          |
> |----------------|----------------|----------------|----------------|----------------|----------------|
> | Riesz s-Energy | **10.59±0.04** | 5.95±0.18      | 10.17±0.04     | 4.80±0.87      | 6.25±0.06      |
> | Das-Dennis     | 10.58±0.04     | **6.16±0.30**  | **10.23±0.12** | **5.21±0.67**  | **6.32±0.13**  |

---

> ### Author Response · Authors · 2025-11-20
> **Response to Reviewer MLZS (2/5)**
>
> ### **Q4:** Have you tried flow models instead of diffusion models?
> - We address this in the newly added Appendix A.5 (Table 5). As established in recent literature (Gao et al., 2024), a diffusion model with a deterministic ODE sampler is mathematically equivalent to Gaussian Flow Matching. We compared our default stochastic SDE sampler against a deterministic ODE sampler (proxy for flow models).
> - The results show that the stochastic sampler outperforms the deterministic one on most benchmarks. We hypothesize that the randomness inherent to the stochastic EDM solver aids exploration, which is particularly beneficial for discovering diverse solutions in offline MOO.
>
> | Method                          | C10/MOP2        | MO-Hopper-v2    | RE34           | Regex           | ZDT2           |
> |---------------------------------|----------------|----------------|----------------|----------------|----------------|
> | Stochastic PCD                  | **10.59±0.04** | _5.95±0.18_    | **10.17±0.04** | 4.80±0.87      | _6.25±0.06_    |
> | Deterministic PCD               | _10.54±0.03_   | 4.47±0.00      | 9.47±0.28      | _5.56±0.48_    | 5.00±0.04      |
> | Stochastic PCD (w/o reweighting)| 10.47±0.01     | **6.01±0.58**  | _10.11±0.03_   | 5.55±0.25      | **7.89±0.19**  |
> | Deterministic PCD (w/o reweighting)| 10.49±0.06  | 4.47±0.00      | 9.24±0.26      | **5.68±0.31**  | 4.98±0.03      |
>
> **Reference:**
>
> Gao, R., Hoogeboom, E., Heek, J., De Bortoli, V., Murphy, K. P., & Salimans, T. (2024). Diffusion Meets Flow Matching: Two Sides of the Same Coin.

---

> ### Author Response · Authors · 2025-11-20
> **Response to Reviewer MLZS (3/5)**
>
> ### **W1:** “Unlike prior methods (Xue et al., 2024; Yuan et al., 2025) that rely on complex, multi-stage pipelines of surrogate predictors or separate optimization algorithms, PCD offers a more direct, end-to-end framework.” In my view, ParetoFlow is classifier guidance while this PCD is classifier-free guidance. I do not think the former is more complicated. Although paretoflow requires two stages: predictor training and generative model training, these are established pipelines for that and thus they should be simple instead of complex.
> - We appreciate the reviewer’s feedback and agree that training separate components is standard practice. We have revised Section 1 to remove the term "complex".
> - However, we highlight that our proposed method does offer benefit in terms of sampling speed, as detailed in the newly added Appendix A.4 (Table 4):
>   - PCD is faster than the generative baseline, ParetoFlow. This is because PCD avoids the computationally expensive and memory-intensive backpropagation required by predictor-guided sampling methods, relying instead on classifier-free guidance.
>
> | Method      | RE34        | MO-Hopper-v2 | Molecule     | C10/MOP2     |
> |-------------|-------------|--------------|--------------|--------------|
> | **Dataset Info** |             |              |              |              |
> | No. of Samples    | 60,000      | 4,500        | 49,001       | 26,316       |
> | No. of Objectives | 3           | 2            | 2            | 3            |
> | No. of Dimensions | 5           | 10,184       | 32           | 31           |
> | **Sampling Time (sec)** |     |              |              |              |
> | PCD         | **1.96±0.13** | **3.22±0.01** | **1.80±0.02** | **2.02±0.14** |
> | ParetoFlow  | 6.08±0.06   | 17.20±0.61   | 36.91±1.99   | 5.81±0.04    |

---

> ### Author Response · Authors · 2025-11-20
> **Response to Reviewer MLZS (4/5)**
>
> ### **Q3:** It looks weird in MORL that no methods can beat against the offline best? So what is the point of all methods?
> - Indeed, we found it quite interesting that no methods were able to outperform the best solution found in the offline dataset. This is likely due to the high dimensionality of the MORL tasks (∼10,000 dimensions) and the relatively small dataset size (approx. 4,500–8,000 points).
>
> - However, we highlight that the MORL tasks serve as a challenging benchmark to evaluate how well the proposed methods scale to very high-dimensional data distributions. This motivates the need for future work in latent-space modeling (e.g., Latent Diffusion) or Transformer-based architectures for neural network parameter generation, which we discuss in our revised limitations section.

---

> ### Author Response · Authors · 2025-11-20
> **Response to Reviewer MLZS (5/5)**
>
> ### **Q5:** I doubt whether we really need diffusion models for some tasks mentioned here. Diffusion models can be very useful if the data has meaningful patterns to learn like image, text, protein. In this context, some designs are pure math numbers. Do they have meaningful patterns?
> - While the real-world utility of synthetic functions is limited, they provide a fast and controllable way to benchmark new methods. We also argue that if the data contained no patterns (e.g., smoothness), no optimization method would be able to generalize effectively.
>
> - Crucially, we highlight that PCD is also evaluated on scientific design tasks (e.g., proteins and molecules). These problems closely resemble real-world settings where diffusion models have proven highly effective at capturing complex data patterns.

---

### Official Review · Reviewer_QMRg · 2025-10-28

**Soundness:** 2
**Presentation:** 4
**Contribution:** 3
**Rating:** 6
**Confidence:** 5

**Summary:**

This paper proposes Pareto-Conditioned Diffusion models (PCD) for offline multi-objective optimization (MOO), which aims to obtain a group of high-performing (i.e., Pareto optimal) final solutions across multiple objectives using only a fixed offline dataset $ \mathcal{D} = \\{ (\boldsymbol{x}_i, \boldsymbol{y}_i) \\} _{i=1}^N$, without any online iterative evalution. While prior works mainly focus on modeling $p(\boldsymbol{y}|\boldsymbol{x})$ and obtaining the inputs that maximize the model’s output, PCD models $p(\boldsymbol{x}|\boldsymbol{y})$ instead via conditioned diffusion models, extending the practice of [1] to the MOO scenarios. Specifically, unlike directly parameterizing the distribution, the authors train the models via binning the objective space $\mathbb{R}^m$, quantified by the dominance number, and assigning weights to each bin, calculated by considering both the number of points inside the bin and performance based on the dominance number. After training, they generate a diverse set of condition $\boldsymbol{y}$ and sample for final solutions via classifier-free guidance. Experimental results demonstrate the superiority of PCD over a wide range of surrogate-based methods.

**Strengths:**

- This paper is well-written and easy to follow.
- The idea of extending forward-surrogate-free (i.e., $\boldsymbol{y} \to \boldsymbol{x}$ instead of $\boldsymbol{x} \to \boldsymbol{y}$) methods to offline MOO is novel. It is still unknown that which of the modelings is better, and the discussion in this paper brings new insights into this field.
- Compared to recent generative-based methods [2-3], PCD does not rely on forward surrogate or classifier, which may introduce compounding error in both the surrogate and the generative model. Instead, PCD employs classifier-free guidance to sample final solution, which is simpler and shows flexible inference control in diffusion model community.

Overall, I think this paper is of high quality. Once my concerns listed below are addressed, I am willing to adjust my ratings.

**Weaknesses:**

- The multi-objective reweighing scheme is somehow heuristic, mainly regarding the choice of dominance number as ranking metric.
    - From my perspective, the dominance number is just a quantification of the Pareto dominance. However, in MOO, there are still other metrics to further comprehensively examine the goodness of the solutions, e.g.,  the Pareto dominance together with crowding distance from NSGA-II [4] (also used in [3]) and the scalarization mechanism from MOEA/D [5]. I suggest comparing some of them.
- Typos:
    - Not correctly using ``\citep{}``, in line 53, 292, 761.
    - line 324: “Baselines.” → “Baseline”

**Questions:**

- Beyond the sampling procedure delivered in Fig. 5,  can PCD recover the dataset distribution by conditioning on different objective space, like the Fig. 4 in [1]? For example, I suggest you examine the modeling performance of PCD by conditioning on different Pareto front layers in the offline dataset (or the test data provided by [6]) and validating if the model can recover the ground-truth points.
- How does the hypervolume change as the sampling timesteps promote?

I can understand that adding experiments on all task from [6] is quite expensive during rebuttal. I would be pleased that you conduct additional experiments on some representative tasks.

## References

[1] Diffusion models for black box optimization. ICML 2023.

[2] ParetoFlow: Guided Flows in Multi-Objective Optimization. ICLR’25.

[3] Preference-Guided Diffusion for Multi-Objective Offline Optimization. NeurIPS’25.

[4] A fast elitist non-dominated sorting genetic algorithm for multi-objective optimization: NSGA-II. IEEE TEvC, 2002.

[5] MOEA/D: A multiobjective evolutionary algorithm based on decomposition. IEEE TEvC, 2007.

[6] Offline multi-objective optimization. ICML 2024.

---

> ### Author Response · Authors · 2025-11-20
> **Response to Reviewer QMRg (1/3)**
>
> We thank the reviewer for their insightful feedback and helpful suggestions to improve our paper. We address the specific questions below.
>
> ### **W1:** The multi-objective reweighing scheme is somehow heuristic, mainly regarding the choice of dominance number as ranking metric. From my perspective, the dominance number is just a quantification of the Pareto dominance. However, in MOO, there are still other metrics to further comprehensively examine the goodness of the solutions, e.g., the Pareto dominance together with crowding distance from NSGA-II [4] (also used in [3]) and the scalarization mechanism from MOEA/D [5]. I suggest comparing some of them.
>
> - We agree that the dominance number captures only one aspect of solution quality. However, our empirical evaluations show that placing higher emphasis on points closer to the non-dominated front improves the method’s performance. We acknowledge that incorporating metrics like crowding distance into the reweighting scheme could help ensure a wider variety of points are considered, and we view this as a promising direction for future work.
>
> - Regarding scalarization (e.g., MOEA/D), we initially explored this idea but found that it resulted in substandard performance compared to our proposed approach. We believe this is due to two main limitations in the offline setting:
>   - Offline data lacks inherent weight vectors, making it unclear how to pick the weights during training.
>   - Standard linear scalarization often fails to capture non-convex Pareto fronts.
>
> - These limitations motivated our decision to condition directly on the targeted trade-offs rather than relying on decomposition.

---

> ### Author Response · Authors · 2025-11-20
> **Response to Reviewer QMRg (2/3)**
>
> ### **Q1:** Beyond the sampling procedure delivered in Fig. 5, can PCD recover the dataset distribution by conditioning on different objective space, like the Fig. 4 in [1]? For example, I suggest you examine the modeling performance of PCD by conditioning on different Pareto front layers in the offline dataset (or the test data provided by [6]) and validating if the model can recover the ground-truth points.
>
> - To address this, we have added a new analysis in Appendix A.7 (Figure 9). We partitioned the training dataset into distinct non-dominated fronts $(F_1, \ldots, F_M)$ and conditioned the model on the objective vectors from these fronts to test if it could recover the corresponding designs. We observe that:
>   - For RE21: The model faithfully reconstructs the ground-truth data distribution, validating that our conditional sampling mechanism works as intended on lower-dimensional engineering design tasks.
>   - For MO-Hopper: The model struggles to recover the specific ground-truth points. This is likely due to the high dimensionality of the task (∼10,000 dimensions) and the relatively small dataset size (4.5K points).

---

> ### Author Response · Authors · 2025-11-20
> **Response to Reviewer QMRg (3/3)**
>
> ### **Q2:** How does the hypervolume change as the sampling timesteps promote?
> - We have investigated this in the newly added Appendix A.6.4 (Figure 7), where we evaluated the Hypervolume across a range of denoising steps (from 32 to 1024). We observe that:
>   - Continuous Tasks (e.g., RE34, ZDT2): Performance is highly robust to the number of steps. Even with as few as 32 steps, the model achieves performance comparable to the default 1024 steps. This is due to our conditioning mechanism: the generated target points are already close to the data distribution, making the denoising task relatively easy.
>   - Discrete Tasks (specifically Regex): Performance drops with fewer steps. This is due to the decoding process being far more susceptible to the small errors caused by fewer sampling steps.

---

> ### Comment · Reviewer_QMRg · 2025-11-20
>
> Thanks for your timely reply. I appreciate your transparency to show off the experimental results (reply to Q1 and Q2) and the design intuition discussed in reply to W1. However, I still think that some experimental supports in W1 are necessary (as the verbal discussion does not make great sense to me) since dominance number is not so prevalent in the community of MOO.
>
> Besides, after reading other reviewers' reviews and your modification in the manuscripts, I understand the authors' great efforts during rebuttal and address some of my concerns. I have raise the soundness score from 2 to 3.

---

> ### Author Response · Authors · 2025-11-25
> **Response to Reviewer QMRg Continued**
>
> We thank the reviewer for raising the question of considering different metrics for evaluating the quality of points for reweighting. We acknowledge that while the dominance number measures the points' ''distance'' from the best set of points, it overlooks other important aspects such as diversity, which can be vital for generating high-quality solutions.
>
> To address these concerns, we conducted additional experiments comparing our dominance number-based reweighting to crowding distance-based reweighting, which emphasizes diversity by placing higher weights on points that are further away from others.
>
> The results, shown in the table below, indicate that our dominance number approach outperforms the crowding distance method in 3 out of 5 tasks. This suggests our approach is effective, yet integrating both dominance number and crowding distance may enhance performance even further. Thus, we see crowding distance as a complementary metric that could be integrated into our proposed reweighting method. We would like to thank the reviewer for pointing out this possible future direction for further improving offline multi-objective optimization.
>
>
> | Task         | Crowding distance   | Dominance number   |
> |:-------------|:------------------------------------------|:-----------------------------------------|
> | C10/MOP2     | $10.55\pm0.05$                            | $\textbf{10.59}\pm\textbf{0.04}$                         |
> | MO-Hopper-v2 | $5.91\pm0.31$                             | $\textbf{5.95}\pm\textbf{0.18}$                            |
> | Molecule     | $3.59\pm0.74$                             | $\textbf{4.07}\pm\textbf{0.51}$                            |
> | RE34         | $\textbf{11.33}\pm\textbf{0.25}$                            | $10.17\pm0.04$                           |
> | Regex        | $\textbf{5.07}\pm\textbf{0.34}$                             | $4.80\pm0.87$                            |

---

> ### Comment · Reviewer_QMRg · 2025-11-25
>
> Thanks again for your response. I have no more concerns. After double checking the manuscript and other reviewers' comments, I think this paper is of soundness & I'm willing to increase my rating now.
>
> Additionally, though not so necessary, I recommend to analyze the performance of PCD over different objectives (e.g., in RE and MO-NAS, it seems that PCD underperforms ParetoFlow on most tasks with #objective greater than 3). What does this come from? It would be interesting to analyze such points.
> > Personally, I guess maybe the classifier guidance? or maybe the scalarization scheme taken be ParetoFlow compared to the dominance number? Since in the MOO community, MOEA/D outperforms dominance-based methods, e.g., NSGA-II, in problems that contain more than 2 objectives.

---

### Official Review · Reviewer_pgCD · 2025-10-28

**Soundness:** 3
**Presentation:** 3
**Contribution:** 3
**Rating:** 6
**Confidence:** 5

**Summary:**

The paper proposes Pareto‑Conditioned Diffusion (PCD), an end‑to‑end framework for offline multi‑objective optimization (MOO) that removes explicit surrogate predictors. PCD trains a conditional diffusion model to sample designs x conditioned directly on target objective trade‑offs y. Two key components make this work in the offline regime: (1) a multi‑objective reweighting of the static dataset that emphasizes bins with high‑quality, non‑dominated samples (using dominance numbers and binning) (2) a reference‑direction mechanism to generate a diverse set of conditioning points by assigning offline points to uniformly spread directions and extrapolating slightly beyond the dataset. The model is then sampled with classifier‑free guidance. On Off‑MOO‑Bench, PCD reports competitive HV.

**Strengths:**

1. Framing offline MOO as conditional sampling is elegant: a single model generates candidate sets conditioned on desired trade‑offs, sidestepping surrogate‑then‑optimizer pipelines.
2. Method components are well‑motivated:
- The reweighting combines dominance‑based quality with bin density, to emphasize promising regions without discarding too much signal
- The reference‑direction procedure provides diverse, high‑quality conditioning targets for sampling
3. The paper evaluates across different task families and runs targeted ablations that isolate each module’s effect; Table 2 shows the reference‑direction mechanism and reweighting both help on several tasks.
4. The authors highlight contradictions with prior works or expectations and give plausible hypotheses. This transparency is commendable.

**Weaknesses:**

1. Computation of dominance numbers. Computing the dominance number o(x) for reweighting is quadratic in dataset size if done naively (Eq. 5). The paper would benefit from complexity notes or empirically evaluating the time cost for sampling with the proposed PCD method.
2. Benchmarks are mostly m≤3 objectives. It’s unclear how the reference‑direction generation and conditioning scale when m grows (e.g., direction coverage, sampling stability, and the number of conditions L required). A small synthetic study with m>3 would help.

**Questions:**

See above

---

> ### Author Response · Authors · 2025-11-20
> **Response to Reviewer pgCD (1/2)**
>
> We thank the reviewer for their insightful feedback and helpful suggestions to improve our paper. We address the specific questions below.
>
> ### **W1:** Computation of dominance numbers. Computing the dominance number o(x) for reweighting is quadratic in dataset size if done naively (Eq. 5). The paper would benefit from complexity notes or empirically evaluating the time cost for sampling with the proposed PCD method.
> - We have included both the preprocessing and sampling costs in Appendix A.4 (Table 4).
>   - Dominance Calculation: While the naive calculation is quadratic, it is a one-time preprocessing step. Using optimized libraries (e.g., pygmo), calculating the dominance numbers for our largest dataset (60,000 samples) takes less than 20 seconds.
>
>   - Sampling Time: As shown in Table 4, PCD is faster than the generative baseline, ParetoFlow. This is because PCD avoids the computationally expensive and memory-intensive backpropagation required by predictor-guided sampling methods, relying instead on classifier-free guidance.
>
> Moreover, our new ablation Appendix A.6.4 (Figure 7) indicates that for continuous tasks, reducing denoising steps from 1024 to 32 has negligible impact on performance, which could lead to faster sampling speeds.
>
> |                  | RE34       | MO-Hopper-v2 | Molecule   | C10/MOP2  |
> |-----------------|-----------|--------------|-----------|-----------|
> | **Dataset Info**        |           |              |           |           |
> | No. of Samples          | 60,000    | 4,500        | 49,001    | 26,316    |
> | No. of Objectives       | 3         | 2            | 2         | 3         |
> | No. of Dimensions       | 5         | 10,184       | 32        | 31        |
> | **Reweighting Time (sec)** | 17.3±0.147 | 0.090±0.000 | 11.8±0.019 | 2.94±0.0125 |
> | **Sampling Time (sec)** |           |              |           |           |
> | Method: PCD             | 1.96±0.13 | 3.22±0.01    | 1.80±0.02 | 2.02±0.14 |
> | Method: ParetoFlow      | 6.08±0.06 | 17.20±0.61   | 36.91±1.99| 5.81±0.04 |

---

> ### Author Response · Authors · 2025-11-20
> **Response to Reviewer pgCD (2/2)**
>
> ### **W2:** Benchmarks are mostly m≤3 objectives. It’s unclear how the reference‑direction generation and conditioning scale when m grows (e.g., direction coverage, sampling stability, and the number of conditions L required). A small synthetic study with m>3 would help.
> - We have conducted the requested scalability study in Appendix A.8 (Table 8) using the synthetic DTLZ1 and DTLZ7 benchmarks with $ m \in \\{3, 4, 5, 6\\} $ objectives.
>   - The results demonstrate that PCD achieves consistent relative hypervolume improvements over the offline dataset regardless of the number of objectives (m).
>   - This highlights the scalability of our reweighting and conditioning mechanisms, demonstrating that PCD effectively generalizes even in high-dimensional objective spaces.
>
> | Task   | 3           | 4           | 5           | 6           |
> |--------|------------|------------|------------|------------|
> | DTLZ1  | 1.29±0.07  | 1.25±0.04  | 1.84±0.16  | 1.37±0.08  |
> | DTLZ7  | 1.33±0.02  | 1.17±0.01  | 1.19±0.02  | 1.21±0.04  |

---

> > ### Comment · Reviewer_pgCD · 2025-11-25
> >
> > Thank you for your response. I appreciate the authors’ efforts in addressing my concerns. I would like to maintain my positive score.

---

### Official Review · Reviewer_mkio · 2025-11-03

**Soundness:** 3
**Presentation:** 3
**Contribution:** 3
**Rating:** 6
**Confidence:** 5

**Summary:**

This paper focuses on offline multi-objective optimization (MOO) and addresses the core challenge of generalizing beyond observed data in static datasets. The motivation stems from existing offline MOO methods, which either rely on error-prone surrogate models or multi-stage pipelines (e.g., ParetoFlow) and struggle to eliminate dependencies on intermediate guidance mechanisms.

The proposed core method, **Pareto-Conditioned Diffusion (PCD)**, reframes offline MOO as a conditional sampling problem using diffusion models. It integrates two key components: a multi-objective reweighting strategy (prioritizing samples near the Pareto front) and an NSGA-III-inspired reference-direction mechanism (generating diverse conditioning points).

Experiments on five benchmark categories (synthetic, MORL, real-world, scientific design, MONAS) show PCD achieves competitive HV scores and outperforms baselines (e.g., ParetoFlow) in cross-task consistency with a single set of untuned hyperparameters. Ablation studies validate its reweighting and reference-direction components.

**Strengths:**

1. PCD eliminates the need for explicit surrogate models or scalarization schemes (common in existing methods like MOBO or ParetoFlow), simplifying the optimization pipeline and reducing risks of exploiting surrogate inaccuracies.
2. The multi-objective reweighting strategy (Equation 6) balances dense bins and high-performance samples, while the reference-direction mechanism (inspired by NSGA-III) ensures diverse, high-quality conditioning points. This design is sound and well-motivated.
3. PCD maintains consistency across tasks with a single set of hyperparameters, and ablation studies confirm the soundness of its mechanism.

**Weaknesses:**

1. PCD underperforms on some challenging tasks, including MORL (high-dimensional) and MONAS (purely categorical). I expect more discussion and guidance for future research. For example, are there targeted modifications (e.g., dimensionality reduction, specialized denoiser architectures) that might address this?
2. The paper explicitly excludes combinatorial tasks (citing needs for specialized denoising) but provides no technical discussion of how PCD could be extended to this subset of MOO.
3. The reference-direction mechanism involves extrapolating points along direction vectors and adding Gaussian noise. Please provide more intuition for key choices (e.g., extrapolation distance, noise variance).
4. The sensitivity analysis (A.2) shows τ must be adjusted for datasets with high quality variance (e.g., ZDT2). How can practitioners **systematically estimate dataset variance** (e.g., via dominance number distribution) and select it without extensive tuning?

**Questions:**

See weaknesses

---

> ### Author Response · Authors · 2025-11-20
> **Response to Reviewer mkio (1/3)**
>
> We thank the reviewer for their insightful feedback and helpful suggestions to improve our paper. We address the specific questions below.
>
> ### **W3:** The reference-direction mechanism involves extrapolating points along direction vectors and adding Gaussian noise. Please provide more intuition for key choices (e.g., extrapolation distance, noise variance).
> - Intuitively, the extrapolation distance pushes the conditioning points further away from the training data manifold, while the noise scale controls the diversity of the generated targets. We have included an ablation of these hyperparameters in the newly added Appendix A.6.2 (Figure 6). We observe that:
>   - Increasing the noise scale produces small but consistent performance gains across tasks.
>   - However, increasing extrapolation distance leads to no improvement in performance.
>
> This confirms that the optimal strategy is to prioritize generating diversity instead of trying to move too far away from the training distribution. Additionally, we have added Figure 2 to the main text to better visualize the overall conditioning point generation procedure.

---

> > ### Author Response · Authors · 2025-11-20
> > **Response to Reviewer mkio (2/3)**
> >
> > ### **W1&2:** PCD underperforms on some challenging tasks, including MORL (high-dimensional) and MONAS (purely categorical). I expect more discussion and guidance for future research. For example, are there targeted modifications (e.g., dimensionality reduction, specialized denoiser architectures) that might address this? The paper explicitly excludes combinatorial tasks (citing needs for specialized denoising) but provides no technical discussion of how PCD could be extended to this subset of MOO.
> > - In the revised Summary and Limitations (Section 6), we have expanded our discussion to provide concrete technical guidance for these challenging domains:
> >   - We suggest addressing the limitations of MLP-based denoisers by adopting Latent Diffusion Models (LDMs) (Rombach et al., 2022) to operate in a compressed latent space or employing Transformer-based architectures (Peebles et al., 2023), both of which have proven scalability in high-dimensional parameter generation (Wang et al., 2024).
> >   - We discuss enhancing PCD using continuous-space diffusion for categorical data (Dieleman et al., 2022), which preserves the applicability of standard classifier-free guidance, or adopting fully discrete diffusion models (Austin et al., 2021).
> >   - We outline a specific technical path for extending PCD to combinatorial tasks by integrating constrained diffusion frameworks (Sun & Yang, 2023) with recent advances in discrete guidance mechanisms (Nisonoff et al., 2025).
> >
> > **References**:
> >
> > Rombach, et al. (2022). High-resolution image synthesis with latent diffusion models.
> >
> > Peebles, et al. (2023). Scalable diffusion models with transformers.
> >
> > Wang, et al. (2024). Neural network diffusion.
> >
> > Austin, J. et al. (2021). Structured Denoising Diffusion Models in Discrete State-Spaces.
> >
> > Dieleman, S. et al. (2022). Continuous diffusion for categorical data.
> >
> > Sun, Y., & Yang, Z. (2023). DIFUSCO: Graph-based diffusion solvers for combinatorial optimization.
> >
> > Nisonoff, D. et al. (2025).  Unlocking guidance for discrete state-space diffusion and flow models.

---

> > > ### Author Response · Authors · 2025-11-20
> > > **Response to Reviewer mkio (3/3)**
> > >
> > > ### **W4:** The sensitivity analysis (A.2) shows τ must be adjusted for datasets with high quality variance (e.g., ZDT2). How can practitioners systematically estimate dataset variance (e.g., via dominance number distribution) and select it without extensive tuning?
> > > - Hyperparameter selection is inherently difficult in the offline setting due to the lack of access to the true objective function for validation. Per the reviewer’s suggestion, we explored systematic ways to select τ, such as deriving it from the standard deviation or Cumulative Distribution Function (CDF) of the dominance numbers. However, we found that these heuristics did not yield consistent improvements across tasks. Despite this, we highlight two key points for practitioners:
> > >   - The default value (τ = 0.05) is robust and achieves competitive and consistent performance on the vast majority of tasks (see Table 1), suggesting that tuning is rarely necessary in practice.
> > >   - For outlier datasets with high variance (like ZDT2), we recommend a simple visual heuristic based on Figure 4: if the distribution of dominance numbers exhibits a "long tail," increasing τ effectively "softens" the reweighting to preserve more training data, as confirmed by our sensitivity analysis.

---

> > > > ### Comment · Reviewer_mkio · 2025-11-28
> > > >
> > > > I have read the authors' responses as well as the other reviewers' comments. I appreciate the authors' efforts during the response process. My concerns have been addressed, and I am pleased to recommend accepting this paper (though I am unable to edit the official comment due to some known issues).

---

### Author Response · Authors · 2025-11-25
**General Response**

We thank all the reviewers for their time and effort in reviewing our work, and especially for their constructive suggestions. In light of these comments, we have made the following changes to improve our manuscript:

* **Scalability and efficiency (Reviewer pgCD):** We added Appendix A.8 to demonstrate performance on DTLZ1 and DTLZ7 tasks with increasing number of objectives $ m \in \\{3, 4, 5, 6\\} $ and Appendix A.4 to analyze computational cost. These results demonstrate effective scaling to many-objective settings and faster sampling times compared to generative baseline ParetoFlow.

* **Discussion on limitations (Reviewer mkio):** We revised Section 6 to provide concrete technical guidance on extending PCD to challenging high-dimensional, discrete, and combinatorial domains.

* **Qualitative analysis and visualization (Reviewers QMRg, mkio):** We added Figure 2 to the main text to visualize the conditioning mechanism and Appendix A.7 (Figure 9) to analyze the model's ability to reconstruct the training distribution.

* **Additional ablations (Reviewers MLZS, QMRg, mkio):** We added comprehensive ablations covering:
    * Stochastic SDE vs. Deterministic ODE Samplers (proxy to Gaussian Flow Matching) (Appendix A.5).
    * Reference direction strategies (Das-Dennis vs. Riesz s-Energy) (Appendix A.6.3).
    * Robustness to denoising steps (Appendix A.6.4).
    * Sensitivity to noise scale and extrapolation distance (Appendix A.6.2).

    The ablation results indicate general robustness to design choices and support the use of the stochastic sampler.

Finally, we fixed typos, cleaned the BibTeX formatting, and moved the LLM usage disclosure from the ethics statement to Appendix A.10. All revisions are highlighted in blue.

---

> ### Author Response · Authors · 2025-12-01
> **Rebuttal Summary**
>
> Dear AC,
>
> Given the shortened discussion period, we provide a summary of our rebuttal to assist your review:
>
> * **Reviewer QMRg (Rating 6 → 8, Confidence 5):** Following a constructive exchange, the reviewer confirmed they have "no more concerns" and raised their rating to 8. We will incorporate their suggestion to include more analysis on many-objective problems in the final version.
>
> * **Reviewers mkio and pgCD (Rating 6, Confidence 5):** We successfully addressed their concerns through additional experiments and detailed responses. Both reviewers confirmed that their concerns were resolved.
>
>   * *Notably, Reviewer mkio stated they are "pleased to recommend accepting", though technical issues prevented them from updating the official score.*
>
> * **Reviewer MLZS (Rating 8, Confidence 4):** We provided a comprehensive response and additional results to address the feedback.
>
> Overall, we believe the revised manuscript has been significantly strengthened during the discussion phase. We sincerely appreciate the reviewers’ positive assessments and look forward to your decision.
>
> Thank you,
>
> The Authors

---

### Public Comment · ~Ruiqing_Sun1 · 2026-06-11
**Critical Issues Regarding Evaluation Validity, Underestimated Baselines, and Reproducibility**

We would like to bring to the authors' attention several issues we encountered while reviewing the paper and running the official code provided in the GitHub repository. These issues significantly affect the validity of the empirical results and the baseline comparisons. We summarize our findings below for the authors' clarification.

**1. Invalid Evaluations due to Violated Decision Variable Constraints**
While evaluating the generated solutions, we found that the proposed model generates decision variables $x$ that fall outside the feasible bounds of the test problems. For standard synthetic tasks like ZDT, the decision variables have strict theoretical boundaries defined within $x \in [0, 1]^d$ (https://pymoo.org/problems/multi/zdt.html).

However, in our reproduction of the ZDT2 task (using `seed=0`), the generated $x$ values ranged from **$-0.847$ to $2.295$**. Feeding unconstrained $x$ values into the objective functions produces incorrect $y$ values, which subsequently inflates the Hypervolume (HV) scores. Our reproduced run yielded `{"hv_d_best": 4.677, "hv_100th": 6.539, "hv_75th": 6.061, "hv_50th": 5.537}`. Without proper boundary constraints (such as clipping or projection), the algorithm operates outside the feasible region, making the reported HV results invalid and incomparable to properly constrained baselines.

After manually truncating x to the range [0, 1], we obtained the following results: `{'hv_100th': 5.06617791444767, 'hv_50th': 4.829375757256848, 'hv_75th': 4.905579386237239, 'hv_d_best': 4.677138547421682}`

We observed similar out-of-bound behaviors across other synthetic functions.

**2. Anomalies in Baseline Evaluation Results**
We noticed systematic discrepancies between the baseline performances reported in this paper and those established in the original Off-MOO-Bench paper (*Xue et al., ICML 2024* https://arxiv.org/abs/2406.03722)  .

*   **Identical scores across different methods:** In Table 9 and Table 10, for tasks like `ZDT1` and `ZDT2`, ten completely different baseline methods (including E2E, MH, MM, MM+COMs, MM+ICT, etc.) with different architectures and regularizations achieved identical scores with zero variance (e.g., exactly **`2.81 ± 0.00`** on ZDT1, **`3.17 ± 0.00`** on ZDT2, and **`0.29 ± 0.00`** on VLMOP1). This phenomenon indicates a likely execution error or crash in the baseline evaluation pipeline (e.g., returning a default state), rather than the actual performance of these methods. In Table 9 (Appendix A.11) for the ZDT1 task, almost all baseline methods (except ParetoFlow) are reported with an HV of less than 3 (e.g., $2.81$), but in other papers like PGD or ICML24, most of baseline are bigger than 4. We found similar phenomenon in other problmes.
*   **Zero and suppressed scores on RE tasks:** In Table 13, the HV scores for nearly all DNN-based baselines (E2E, MH, MM) on Real-world Engineering tasks `RE33`, `RE35`, `RE36`, and `RE42` are reported as exactly **`0.00 ± 0.00`**. However, the original ICML 2024 paper shows these baselines achieving competitive non-zero performances (e.g., E2E+GN achieving $> 11.5$ on RE35, $> 50.0$ on RE34, and $> 140.0$ on RE61).


**3. Omission of the PGD-MOO Baseline**
In Footnote 2 (Page 8), the paper states that PGD-MOO was omitted because the authors "were unable to obtain results on our broader benchmarks using the official code." We tested the official repository of PGD-MOO and were able to successfully run their code and reproduce results on these benchmark tasks. Given that PGD-MOO is a highly relevant generative baseline for offline MOO, it should be included in the comparison.

**4. Exclusion of DTLZ2-DTLZ6**
In Footnote 1 (Page 6), tasks DTLZ2-DTLZ6 are excluded due to an evaluation error in the original benchmark, citing GitHub issue #14. We checked the repository and noted that this specific issue was officially patched and closed on March 21, 2025. Since the bug was resolved long before the submission deadline, excluding these standard multi-objective test functions reduces the comprehensiveness of the evaluation.

---

> ### Public Comment · ~Jatan_Shrestha1 · 2026-06-17
>
> **1.** Thank you for bringing this to our attention. Our pipeline relies on the task oracle to handle design feasibility, and due to an oversight on our part, we were not aware that the synthetic oracles (e.g. the ZDT and VLMOP functions) inherited from the ParetoFlow codebase (offline MOO benchmark) do not enforce boundary clipping before evaluation. We will fix this and re-run PCD and all baselines including ParetoFlow under the corrected evaluation, updating the affected results and their corresponding tables.
>
> **2.** As noted in the paper, we attribute the deviations observed in some baselines to the updated datasets released by the Offline MOO authors. We reached out to the authors and they confirmed that the newer datasets differ from the original ones but had not re-benchmarked on them. We reproduced the baselines using their codebase and default hyperparameters as faithfully as possible. While the performance of baselines in RE tasks differs significantly from what was reported in [1], these results were verified multiple times using the default hyperparameter configurations.
>
> We would like to highlight that the ***tasks in which the baselines obtained a hypervolume of zero were excluded from the computation of average ranks (Table 1 in our paper), thus providing a fair representation of our proposed solution.*** It should be noted that both PGD-MOO [2] and ParetoFlow [3] present results that differ from each other, and from the original paper [1], indicating that these issues are not isolated to our efforts, but rather part of a broader reproducibility issue with the benchmark itself.
>
> **3 & 4.** We built our framework on the ParetoFlow codebase to run the MO-NAS benchmarks, which were not running in the main Off-MOO-Bench repository. As a result, our synthetic suite follows the ParetoFlow setup, which evaluates only DTLZ1 and DTLZ7, consistent with the known evaluation error in the original benchmark that we cite in the paper. Nevertheless, we agree that inclusion of DTLZ2-DTLZ6 could improve the coverage of our synthetic benchmark suite.
>
> Finally, regarding PGD-MOO: at the time of our evaluation, the PGD-MOO preprint only covered Synthetic and RE benchmarks. While we contacted the authors of PGD-MOO to get access to a more up-to-date codebase, we did not receive it in time for the submission. Regardless, we agree that PGD-MOO is a relevant method, and many of our proposed improvements (such as dataset reweighting) could also be extended to PGD-MOO.
>
> We appreciate these comments and are happy to clarify further if anything remains unclear!
>
> [1] Offline Multi-Objective Optimization. (ICML 2024)
>
> [2] Preference-Guided Diffusion for Multi-Objective Offline Optimization. (NeurIPS 2025)
>
> [3] ParetoFlow: Guided Flows in Multi-Objective Optimization. (ICLR 2025)

---

### Meta-Review · Area_Chair_Tn2S · 2026-01-06

**Summary:**

This paper studies offline multi objective optimization and addresses the challenge of generalizing beyond the observed data in a given dataset. To this end, the authors formulate offline multi objective optimization as a conditional sampling problem and propose a diffusion based approach. The method incorporates two key components: a multi objective reweighting strategy and a reference direction based mechanism to encourage diverse conditioning points. Experimental results demonstrate the effectiveness of the proposed approach and, importantly, show strong generalization across multiple tasks using a single set of hyperparameters.

**Reviewer Concerns:**

All reviewers are positive about this work. They agree that formulating multi objective optimization as a conditional sampling problem is elegant and promising, and effectively addresses the limitations of existing surrogate based methods and multi stage pipelines. In addition, the idea of searching the Pareto frontier and then generating diverse solutions via conditional diffusion models is considered novel and interesting. Most reviewer comments focus on clarification, and the authors have addressed these questions adequately in their responses. After the discussion phase, one reviewer increased their score from 6 to 8.

I have also read the paper carefully and strongly agree with the reviewers’ assessment that this work provides an interesting and meaningful contribution to multi objective optimization, beyond existing surrogate methods and recent approaches such as ParetoFlow. Overall, this is a solid piece of work, and I am inclined to accept it.

**Reviewer Scores:**

The reviewers’ concerns have been satisfactorily addressed. One reviewer indicated an intention to increase their score, and the current scores after discussion are 8, 8, 6, and 6, which reflect the overall strengths of the paper.

---

### Decision · Program_Chairs · 2026-01-26

Accept (Oral)